# Phytochemicals as Antimicrobials: Prospecting Himalayan Medicinal Plants as Source of Alternate Medicine to Combat Antimicrobial Resistance

**DOI:** 10.3390/ph16060881

**Published:** 2023-06-15

**Authors:** Mohammad Vikas Ashraf, Shreekar Pant, M. A. Hannan Khan, Ali Asghar Shah, Sazada Siddiqui, Mouna Jeridi, Heba Waheeb Saeed Alhamdi, Shoeb Ahmad

**Affiliations:** 1Department of Biotechnology, School of Biosciences and Biotechnology, Baba Ghulam Shah Badshah University, Rajouri 185 234, India; vikasashraf@bgsbu.ac.in; 2Centre for Biodiversity Studies, School of Biosciences and Biotechnology, Baba Ghulam Shah Badshah University, Rajouri 185 234, India; shreekarpant@bgsbu.ac.in; 3Department of Zoology, School of Biosciences and Biotechnology, Baba Ghulam Shah Badshah University, Rajouri 185 234, India; drmahkhan@bgsbu.ac.in (M.A.H.K.); aashah@bgsbu.ac.in (A.A.S.); 4Department of Biology, College of Science, King Khalid University, Abha 61413, Saudi Arabia; mjoridi@kku.edu.sa (M.J.); halhamdi@kku.edu.sa (H.W.S.A.)

**Keywords:** antimicrobial resistance, antimicrobials, multidrug resistance, phytochemicals, phytocompounds, plant secondary metabolites

## Abstract

Among all available antimicrobials, antibiotics hold a prime position in the treatment of infectious diseases. However, the emergence of antimicrobial resistance (AMR) has posed a serious threat to the effectiveness of antibiotics, resulting in increased morbidity, mortality, and escalation in healthcare costs causing a global health crisis. The overuse and misuse of antibiotics in global healthcare setups have accelerated the development and spread of AMR, leading to the emergence of multidrug-resistant (MDR) pathogens, which further limits treatment options. This creates a critical need to explore alternative approaches to combat bacterial infections. Phytochemicals have gained attention as a potential source of alternative medicine to address the challenge of AMR. Phytochemicals are structurally and functionally diverse and have multitarget antimicrobial effects, disrupting essential cellular activities. Given the promising results of plant-based antimicrobials, coupled with the slow discovery of novel antibiotics, it has become highly imperative to explore the vast repository of phytocompounds to overcome the looming catastrophe of AMR. This review summarizes the emergence of AMR towards existing antibiotics and potent phytochemicals having antimicrobial activities, along with a comprehensive overview of 123 Himalayan medicinal plants reported to possess antimicrobial phytocompounds, thus compiling the existing information that will help researchers in the exploration of phytochemicals to combat AMR.

## 1. Introduction

Among the antimicrobials available for medication to clinicians globally, antibiotics hold the prime position. The history of antibiotics dates back to 1904, when the first antibiotic, *Arsphenamine*, was discovered and commercialized as “*Salvarsan*” in 1910, which was used to treat syphilis [1,2]. This was followed by the discovery of sulphanilamide precursor prontosil red in 1927, which was active against *Streptococci* and *Staphylococci*. However, the accidental discovery of penicillin in 1928 from *Penicillium notatum* sparked a quest for antibiotics derived from microbes, leading to the discoveries of streptomycin in 1944 from *Streptomyces griseus*, tetracycline from *Streptomyces rimosus*, and chloramphenicol from *Streptomyces venezuelae* by the end of 1953 [3,4]. Although the search for and discovery of antibiotics started earlier in the 1900s, however, the period from 1940 to 1962 is considered the “golden era” of antibiotic discovery because most antibiotics were discovered in that period, and those antibiotics have significantly contributed to the general health of humans via infectious disease modulation [3,4,5].

The usage of antibiotics proved extremely beneficial for treating infectious diseases, which were otherwise very difficult to cure, but it also exposed microbes to toxic conditions imposed by antibiotics, thus generating a selection pressure, which microbes must overcome to survive [6]. In response to this newly introduced selection pressure, microbes evolved mechanisms to circumvent the lethal effects of antibiotics. This selection pressure grew further due to extensive and improper use of antibiotics, which resulted in the emergence of microbes resistant to these antimicrobials [6,7]. The emergence of antimicrobial resistance (AMR) further compelled the usage of higher dosages or multiple combinations of antibiotics for treatment, all of which results in enhanced usage of antibiotics, thus further accelerating the pace of AMR emergence in bacteria [8,9]. Bacteria evolve comparatively faster than other organisms, primarily due to the small genome size, fast reproduction cycle, and a high rate of development of mutants, resulting in the evolution of resistant phenotypes [10,11].

Almost all antibiotics currently available for medication were discovered in the last century, while those introduced after the year 2000 are only the members of previously discovered antibiotic classes. However, resistance has emerged against almost all of these antibiotics [3]. The emergence of AMR against existing antibiotics is one of the major contributors to worldwide mortalities [4]. According to a review commissioned by the UK government, AMR could kill 10 million people per year by 2050 [12]. This has necessitated the introduction of novel antimicrobials, which has significantly slowed down in the last few decades and thus has created a void in the options available for the treatment of infectious diseases, along with the exploration of alternative sources of medicine.

Plant-based medicines are natural lucrative alternatives, as during the pre-antibiotic era, herbal medicines were extensively used for the treatment of various diseases. Plants, being easily available, were always a choice for the treatment of infectious diseases. Plants produce secondary metabolites for defensive purposes, many of which have antimicrobial properties, and are still commonly used in traditional medicine [10]. Further, plants produce a plethora of structurally and functionally diverse phytochemicals, many of which are effective against pathogenic microbes and hence could be explored for developing novel antimicrobials [13,14,15]. Systematic screening of phytochemical reservoirs of medicinal plants, known to possess antimicrobial properties, could lead to the identification of novel antimicrobial phytochemicals with unique mechanisms of action. Such novel phytochemicals could act by killing bacteria on their own, by inhibiting their molecular targets within the cell, necessary for cellular growth and division, or by acting synergistically with existing antibiotics by inhibiting the resistance determinants in antimicrobial-resistant bacteria. In both cases, the goal of reviving antimicrobial therapies and sensitizing the AMR bacteria could be achieved.

In India, around 17,000 species of higher plants have been discovered, of which 7500 plant species have been found to have medicinal properties [16,17]. The Indian Himalayas foster around 10,000 species of higher plants, of which 1748 species reportedly have medicinal properties [18,19]. Though vast literature is available citing the antimicrobial activity of Himalayan medicinal plants, reports describing their associated bioactive compounds and their molecular mechanisms of action are limited. In an attempt to bridge this gap, the present study focuses on summarizing the emergence of and mechanisms employed by microbes to gain antimicrobial resistance against existing antibiotics, followed by the description of potent phytochemicals, such as alkaloids, phenols, organosulfur compounds, and terpenes, that possess antimicrobial activities, along with their reported mechanisms of action. Finally, this study provides a comprehensive overview of 123 Himalayan medicinal plants having antimicrobial properties along with their bioactive compounds and target pathogens. This review provides an overall picture of the present state of antibiotic discovery and the emergence of antimicrobial resistance, along with the potential of phytochemicals possessing antimicrobial activity, which could be harnessed to screen and develop novel antibacterials that complement present antibiotic therapy by acting as a source of alternate medicine.

### Emergence of Antimicrobial Resistance

Numerous microbe-derived antimicrobials have been bought to the market since the introduction of antibiotics in 1930. However, extensive and inappropriate usage of antibiotics has imposed additional selection pressure, under which microbes have evolved to engage various strategies to resist the effects of antibiotics. Although evolution is a natural process, the wide usage of antibiotic regimes, further compounded by overuse, has accelerated the emergence of AMR in bacteria, which has proven to be a serious problem for the treatment of infectious diseases all around the world [6]. The first penicillin-hydrolyzing enzyme (β-lactamases) was discovered by Chain and Abraham in 1940, soon after the discovery of the first β-lactam antibiotic, penicillin, in 1928 [20]. Similarly, streptomycin, an aminoglycoside antibiotic, was discovered in 1943, and by 1946, resistance against it was reported in bacteria such as *Mycobacterium tuberculosis* [21]. The first tetracycline antibiotic, aureomycin, was discovered in 1945, and resistance against it was observed in *Escherichia coli* by 1948 [22]. Chloramphenicol was introduced in 1947, but resistance against it emerged in the microbial world by 1950 [23]. To cope with resistance against penicillin, methicillin, a semi-synthetic penicillin derivative, was introduced in 1959, but soon after, methicillin-resistant *Staphylococcus aureus* (MRSA) emerged in 1961 [24]. A summary of antibiotic discovery followed by the emergence of antimicrobial resistance is depicted in Figure 1.

It is evident from the examples cited above that the introduction of novel antibiotics was always followed by the emergence of resistance against them. This strongly suggests that the usage of antibiotics itself is responsible for the emergence of antimicrobial resistance among bacteria [25]. The widespread use of antibiotic regimens worldwide has significantly added to the selection pressure during natural bacterial evolution and has accelerated the emergence of resistant strains. Bacteria thrive in harsh environments, reproduce quickly, and have small genomes, which allows them to evolve comparatively faster than other organisms due to faster accumulation of mutations and, thus, acquisition of new phenotypes [10]. In bacteria, the acquisition of an antimicrobial-resistant phenotype involves both intrinsic and extrinsic factors, which include the development and accumulation of de novo mutations in existing genes, resulting in altered phenotypes, as well as the acquisition of resistance-conferring genes through vertical or horizontal gene transfer from other resistant strains [10,11,26]. The most common mechanisms through which bacteria demonstrate resistance involve increased activity of efflux pumps, enzymatic modification or degradation of antibiotics, modifications of target sites, and alteration of membrane permeability through porin modifications [10,25]. These mechanisms are briefly discussed in the following sections.

## 2. Mechanisms of Antimicrobial Resistance

### 2.1. Bacterial Efflux Pumps

Although bacteria can employ different adaptive mechanisms to develop resistance against antimicrobials, efflux pumps hold key importance [27]. Recent data and laboratory studies have demonstrated that efflux pumps not only contribute to AMR but also play a substantial role in microbial adaptive potential and virulence [28]. Bacterial efflux pumps commonly belong to one of two major superfamilies: the widely prevalent secondary transporters that use proton motive force (PMF) as a source of energy and ATP-binding cassette (ABC) multidrug transporters [27,29]. The first superfamily comprises four subfamilies. These include multidrug and toxic compound extrusion (MATE), the major facilitator superfamily (MFS), resistance–nodulation–cell division (RND), and the small multidrug resistance (SMR) family [10,27,30]. Although efflux pumps from all the superfamilies and subfamilies enable a bacterial strain to exhibit increased AMR, RND pumps in particular show activity against a variety of compounds with diverse chemical structures, including bile salts, detergents, organic solvents, antimicrobial peptides, biocides, detergents, and organic solvents [31]. Gram-negative bacteria exhibit complex efflux pumps, which form a tripartite protein channel comprising a transporter protein found in the cytoplasmic membrane, an efflux protein in the outer membrane, and a membrane fusion protein that travels through the periplasm. AcrA-AcrB-TolC of *Escherichia coli* and MexA-MexB-OprM of *Pseudomonas aeruginosa* are two of the most well-studied tripartite systems [28]. Gram-positive bacteria have relatively simple efflux pumps comprising a single transporter, embedded in cytoplasmic membranes, and belong to the ABC, MFS, or SMR families [10,32]. Table 1 summarizes various efflux pumps belonging to ESKAPE pathogens, which include *Enterococcus faecium*, *Staphylococcus aureus*, *Klebsiella pneumoniae*, *Acinetobacter baumannii*, *Pseudomonas aeruginosa*, and members of *Enterobacteriaceae*. As a whole, these efflux pumps enable bacteria to lower the concentration of various antimicrobials inside the cell, to the point that is not lethal to the bacteria.

### 2.2. Enzymatic Modification and Degradation of Antibiotics

Ever since the introduction of antimicrobials for the treatment of infectious diseases, resistance against these compounds had also emerged, which limits the options available for the treatment of disease. Resistance can be active, which results from selection pressure against a specific antibiotic/class of antibiotic, or passive, which results from a generalized adaptive process [55]. Production of enzymes, which changes the chemical structure of an antibiotic and renders it ineffective, is one of the most crucial strategies for developing resistance to antibiotics. Both Gram-positive as well as Gram-negative bacteria produce enzymes that bring about chemical alteration or modification of antibiotics. The most frequently encountered alteration mechanism involves group transfers, such as adenylation, acetylation, and phosphorylation, brought about by transferases. Moreover, bacteria also produce enzymes that specifically break down antibiotics. For instance, many pathogens secrete β-lactamases, which degrade antibiotics with β-lactam rings by specifically hydrolysing their amide bonds, rendering the antibiotic ineffective [56,57]. The following section discusses these enzyme classes in detail, which are listed in Table 2.

#### 2.2.1. Hydrolysis

Several antibiotics possess chemical bonds that are vital for their activity and surprisingly hydrolytically susceptible, e.g., esters and amides, which are susceptible to enzymes that cleave these chemical bonds, rendering the antibiotics inactive. Among these enzymes, amidases, specifically those referred to as β-lactamases, hydrolyse the amide bond of the β-lactam ring present in penicillin, cephalosporin, and carbapenem classes of antibiotics. Other enzymes include esterases and epoxidases, which cleave the macrolide lactone rings and fosfomycin oxirane rings, respectively [55].

##### β-Lactamases

β-Lactamases constitute an enzyme superfamily, with around 2000 members, and hydrolyse the amide linkage of the β-lactam ring, which is a common structural and functional moiety of all β-lactam antibiotics [58]. β-Lactamases are categorized into four molecular classes (A, B, C, and D), based on amino acid sequence homology [59]. Class B constitutes metalloenzymes, while classes A, C, and D enzymes are serine hydrolases. β-lactamases of class A (such as *CTX-M*, *KPC*, *SHV*, and *TEM*) are the most common ones found in and around human settlements [60,61]. Among class A β-lactamases, TEM and SHV are more prone to mutational variability [61]. They can hydrolyse members of second-to-fourth-generation cephalosporins due to crucial mutations in the active site [62]. These mutational variants are referred to as extended-spectrum β-lactamases (ESBLs) [63]. β-Lactamases belonging to class C effectively hydrolyse cephalosporins. Primarily, the members of this class were encoded by chromosomally located genes, with inducible expression. However, later, these genes were also found to be located on mobile genetic elements [64]. β-lactamases of class D include OXA-type carbapenemases, with the highest structural diversity among all serine hydrolases. Finally, class B comprises all metallo-β-lactamases (MBLs), having one or two zinc ions in their active site, which are required for catalysis [63,65]. Except for monobactams, MBLs hydrolyse practically all β-lactam antibiotics and are inhibited by EDTA, dipicolinic acid, and phenanthroline, which are metal-ion-chelating agents.

A dedicated database of β-lactamases containing the models generated, their classification, and their characterization along with the associated available literature has been developed and maintained at www.bldb.eu (Accessed on 15 November 2022) [66]. Figure 2 elaborates on the molecular and functional classification, substrate specificity, and inhibition profiles of β-lactamases.

##### Macrolide Esterases

Esterases are responsible for the development of resistance to macrolide antibiotics, which catalyse the hydrolysis of macrolide lactone rings [67,68]. However, the 16-membered macrolides such as spiramycin and tylosin are not their preferred substrates. *EreA* and *EreB* are erythromycin esterases of crucial clinical relevance. Compared to *EreA*, *EreB* has a wider substrate profile. Except for telithromycin, *EreB* confers resistance to practically all 14–15-membered macrolides, including roxithromycin and azithromycin. The genes encoding macrolide esterases are located on the plasmids in association with other antibiotic resistance determinants [67,69].

##### Epoxidases

Resistance to antibiotics, such as fosfomycin, is caused due to enzymatic opening of an epoxide ring mediated by a thiol-containing co-substrate or water. The presence of orthologues of this enzyme’s gene in bacterial chromosomes suggests that resistance behaviour due to epoxidases may be common in the environment [63].

### 2.3. Group Transfer

Transferases are the most varied and hence the largest class of resistance-conferring enzymes. They chemically modify antimicrobial drugs by transferring different chemical groups through covalent modification, thus altering the physical properties of the drug [55,70,71]. The members of different groups differ in terms of specificity towards different substrates, types of group transfer, and catalytic mechanisms. The chemical strategies employed by the enzymes to modify antibiotics include N- and O-acylation (aminoglycoside, chloramphenicol, and type A streptogramins), O-phosphorylation (aminoglycoside, macrolide, rifamycin, and peptides), O-nucleotidylation (aminoglycoside and lincosamides), O-ribosylation (rifamycin), O-glycosylation (macrolide and rifamycin), and transfer of thiol (fosfomycin). To carry these reactions, a co-substrate such as ATP, acetyl-CoA, NAD+, UDP glucose, or glutathione is required, which acts as a group donor, to bring about covalent modifications. As a result, these enzymes are predominantly active in the cytosol [55,70].

### 2.4. Miscellaneous Mechanisms of Antibiotic Degradation

#### 2.4.1. Redox Enzymes

As the name indicates, redox enzymes oxidise or reduce antibiotics. Among the most well-studied examples is TetX-catalysed tetracycline oxidation [72,73]. Though TetX is a flavoprotein that requires oxygen for function, paradoxically, a *tetX* gene was found on a plasmid in obligate anaerobe bacteria. Therefore, tetracycline resistance was discovered phenotypically only when *tetX* was cloned into *Escherichia coli* [72,74]. TetX causes tetracycline monohydroxylation, which destroys the metal ion (Mg^2+^) binding site necessary for its antibacterial activity.

#### 2.4.2. Lyases

Lyases are enzymes that cause non-oxidative or non-hydrolytic cleavage of carbon–carbon, carbon–sulfur, carbon–oxygen, and carbon–nitrogen bonds. The best-studied enzyme that brings about this cleavage is virginiamycin B lyase (Vgb), responsible for the resistance of type B streptogramins [75]. The VgB lyase was first cloned from streptogramin-resistant *Staphylococci*. Streptogramins are hexadepsipeptides or heptadepsipeptides, cyclized through an ester bond. The enzyme Vgb catalyses the lysis of an ester bond, leading to the opening of the antibiotic ring, rendering it ineffective [75,76].

### 2.5. Target Site Modification

One of the most common and generalized mechanisms employed by bacteria to gain resistance against antibiotics is modification of their binding sites, which is effective for almost all antibiotic classes. The target alterations involve modification of binding sites by enzymatic addition of chemical groups (e.g., methyl groups), modification of the target site by introducing point mutations in the gene itself, and replacement of the original target by an altered one [26].

#### 2.5.1. Enzymatic Alteration of Target Site

One of the best examples of enzymatic target site modification is the methylation of ribosomes, which is caused by an enzyme that is expressed by the erythromycin ribosomal methylases (*erm*) gene, resulting in the emergence of macrolide resistance [69]. The addition of 1 or 2 methyl groups to an adenine residue (A2058) of 23S rRNA belonging to the 50S ribosomal unit brings about biochemical changes that impair the binding of antimicrobial agents to their target sites [26,77]. The *erm* gene is reported to confer resistance to what is known as the MLSB (macrolides, lincosamides, and streptogramin B) group of antimicrobials. The underlying reason for this cross-resistance is the overlapping binding sites of these antibiotics in the 23S rRNA [26,78,79].

#### 2.5.2. Replacement or Bypass of Antimicrobial Binding Site

The acquisition of methicillin resistance by *Staphylococcus aureus* (MRSA) is a classic example of this mechanism. Penicillin-binding proteins (PBPs) are essential for the transpeptidation and transglycosylation of peptidoglycan units, to which β-lactam antibiotics bind, causing their inhibition and thus preventing bacterial cell wall synthesis. Methicillin resistance is developed due to the acquisition of a *mecA gene*, often found to be located on staphylococcal chromosomal cassette mec (SCC mec), which codes PBP2a, an altered variant of PBP that has low affinity for all β-lactam antibiotics, including penicillins, cephalosporins (all generations except fourth and fifth generation), and carbapenems [80].

Similar to β-lactam antibiotics, glycopeptides (vancomycin and teicoplanin) also inhibit bacterial cell wall biosynthesis. In contrast to β-lactams, glycopeptides attach to the acyl-_D_-Ala-_D_-Ala residues of the developing cell wall rather than PBPs, thus blocking PBP- mediated cross-linking of peptidoglycan and subsequently inhibiting cell wall production [77]. Resistance to vancomycin is particularly prevalent in *Enterococci* (especially *Enterococcus faecium*) and is generally mediated through the acquisition of *van* gene clusters. Genes in these clusters encode enzymes, which modify the synthesis of peptidoglycans through two routes: (i) replacement of final _D_-Ala of the polypeptide with either _D_-lactate or _D_-serine and (ii) destruction of “regular” _D_-Ala-_D_-Ala ending precursors, thus preventing interaction of vancomycin with the precursors of cell wall [81].

#### 2.5.3. Mutations in the Genes Encoding Target Sites

Resistance to rifamycin is a suitable example of mutational resistance. Rifamycin binds to bacterial DNA-dependent RNA polymerase, thus blocking transcription. The binding site of rifamycin is located in a pocket in the β-subunit of RNA polymerase, encoded by the *rpoB* gene. Once bound, the antibiotic hinders transcription by blocking the path of newly synthesized RNA [82]. Resistance to rifamycin emerged due to mutation in the *rpoB* gene, resulting in single-amino-acid substitution, which not only decreased the affinity of the antibiotic towards its target site but also did not affect the catalytic proficiency of the polymerase [83].

### 2.6. Porin Modification

Gram-negative bacteria exhibit resistance to antibiotics primarily due to the presence of an outer membrane, which acts as a permeability barrier and confers intrinsic resistance to particular antibiotic drugs. Porins are the main sites of entrance for several hydrophilic antimicrobials including β-lactams, chloramphenicol, fluoroquinolones, and tetracyclines to cross the bacterial outer membrane. However, any change in the permeability of porins leads to acquired resistance against antibiotics that were previously effective [25]. Decreased porin expression or any mutation altering their structure or function can contribute to the development of acquired resistance. Changes in porin expression typically result in low-level antibiotic resistance. However, the coexistence of other resistance determinants, combined with the changes in porin expression, has been reported to enhance the level of resistance [25]. In essence, the effect of low porin expression resulting in less antibiotic intake, coupled with the action of already-existing resistance mechanisms such as efflux pumps or antibiotic-degrading enzymes, results in a high level of resistance [25].

## 3. Antimicrobial Resistance Modulation via Natural Products

Infectious diseases constitute one of the major contributing factors towards high mortalities worldwide, and the slow discovery of novel antibiotics has created a void in the available treatment options, which has necessitated the need to revisit and explore natural resources [84]. In the pre-antibiotic era, since people were completely dependent on natural resources for all their needs, including medicines, these resources significantly contributed to the treatment of various diseases. Among natural products, microbes and plant products hold prime importance. Since the discovery rate of microbially derived antimicrobials is at its lowest since the golden age of antimicrobials and the emergence of AMR has further narrowed down the treatment options, exploration of alternative medicine based on plant products has become necessary [85]. Plants, being easily available and easy to handle, were the first to be used as treatment options for infectious diseases, which continues even today in many tribal communities as an alternative to modern antibiotics. Plants produce hundreds and thousands of structurally and functionally diverse phytochemicals that exert a multitargeted impact on pathogenic microbes, ensuring their death and no further resistance development [13,14,85,86]. Given the availability of huge phytochemical reserves in the plant kingdom, exploring them for antimicrobial agents seems promising.

### 3.1. Multiple-Compound Synergy vs. Single-Compound Therapy

Plants, as living organisms, are complex systems that are self-organising and environmentally adaptive. These complex adaptive traits are a function of the complex chemical matrix that works in synergy to give rise to complex systems such as plants [13]. Plants thrive in diverse habitats, which are vulnerable to pathogenic attacks, and unlike animals, plants do not possess an adaptive immune system. Therefore, they produce structurally and functionally diverse chemicals known as plant secondary metabolites (PSM), which are functionally so diverse that they not only kill pathogenic microbes but also ensure that there is no resistance development anytime soon [87,88,89]. For instance, a recent study conducted on *Artemisia annua* L. crude extracts (herbal tea) and pure artemisinin resulted in a 6–18-fold reduction in plasmodial IC_50_ in the case of crude extract as compared to purified artemisinin [90]. This phenomenon is explained by the existence of interacting and potentiating compounds in the crude extract that enhance its activity in comparison to the single active compound [87,88,89]. Further insights are needed to decipher the interaction of phytocompounds in a mixture to devise efficient antimicrobials from plant secondary metabolites.

### 3.2. Plant Secondary Metabolites as Antimicrobials

Since the advent of antibiotics in 1930, many classes of microbe-derived antimicrobials have been introduced in the antibiotics market. However, over time, cross-resistance to these antibiotics has proved to be a grave issue for infection treatment worldwide, particularly in developing countries [7,8]. Over-the-counter availability and ignorant consumption of antibiotics have significantly contributed to the evolution of multidrug resistance among microbes [7,25].

With the growing antimicrobial resistance in microbes, recent years have shown a great shift towards alternative therapies, compared to conventional antibiotics, including increasing use of natural products. There has been a growing need for the use of alternate therapies, especially those derived from plants [91,92,93,94]. Plant metabolites are being used directly or as precursors for new synthetic products [95]. Due to their having almost no side effects, most people worldwide prefer biological components for maintaining their health [96]. The first phytocompound used as medicine was morphine, which was discovered from opium poppy (*Papaver somniferum* L.) [97]. Since then, chemicals found in plants that have the potential to treat disease have been widely used, usually in crude forms. However, the period after the 1980s saw a dramatic shift in pharmaceutical firms towards synthetic chemistry or, most appropriately, towards combinatorial chemistry for more efficient and economical drug development options [98,99]. The effectiveness of plant secondary metabolites as herbal formulations and antimicrobial agents has prioritized the use of phytocompounds in drug development against multidrug-resistant microbes [15,100,101]. However, despite extensive research, the Food and Drug Administration (FDA) has authorised only a few phytochemicals, such as capsaicin, codeine, paclitaxel, reserpine, and colchicine, as antimicrobial agents against drug-resistant microbes [10,57]. Crude methanolic extracts of several plants such as lemongrass, neem, *Aloe vera* L., oregano, rosemary, thyme, and tulsi have demonstrated effective antimicrobial activities, which were attributed to the presence of flavonoids and tannins in their crude extracts [102].

Apart from crude extracts, which represent the synergistically cooperating mixture of various phytochemicals, several active formulae in their purified states have shown activity against MDR pathogens along with their molecular targets. To name a few, baicalein found in *Thymus vulgaris* L., *Scutellaria baicalensis* Georgi, and *Scutellaria lateriflora* L. has shown antimicrobial activity against MRSA, which can be attributed to its potential to inhibit the NorA efflux pump of MRSA [103,104]. Berberine, isolated from *Berberis* L. sp., has demonstrated antimicrobial activity by inhibiting bacterial gyrase/topoisomerase, RNA polymerase, and cell division [105]. Magnolol, isolated from the bark of *Magnolia officinalis* Rehder & E.H.Wilson, has demonstrated synergistic activity with meropenem by inhibiting New Delhi metallo-β-lactamase (NDM-1), thereby restoring its activity against NDM-1 expressing *Escherichia coli* [106]. Plasmid-mediated antimicrobial resistance is one of the underlying reasons for bacteria exhibiting resistance behaviour in response to antibiotics. Several phytochemicals, such as 8-epidiosbulbin-E-acetate isolated from *Dioscorea bulbifera* Russ. ex Wall., have been reported to possess curing efficiency against resistance plasmids of *Enterococcus faecalis*, *Shigella sonnei*, *Pseudomonas aeruginosa*, and *Escherichia coli* [107].

Given the promising results of plant-based bioactive compounds against antibiotic-resistant strains and the slow discovery of new and efficient antibiotics, it has become highly imperative to explore the vast repository of phytocompounds to overcome the looming catastrophe of antimicrobial resistance. The antimicrobial activities and mechanisms of action hence employed by phytochemicals such as alkaloids, flavonoids, organosulfur compounds, and terpenes are discussed in the following sections, and a summary is depicted in Figure 3.

#### 3.2.1. Alkaloids

Alkaloids are heterocyclic nitrogenous compounds, biosynthetically derived from amino acids, and show variability in chemical structures [108,109]. The activity of alkaloids against microbial infections is mainly attributed to their inhibitory effects against efflux pumps. Many alkaloid compounds have been reported to have marked significance in the treatment of microbial infections.

Berberine is an isoquinoline alkaloid found in the bark of the stem and roots of *Berberis* L. species and is found to possess antimicrobial activity against various microbes including bacteria, fungi, protozoa, and viruses [92]. The mode of antimicrobial activity of berberine is attributed to DNA intercalation, inhibition of RNA polymerase, and inhibition of DNA gyrase and Topoisomerase IV [57,110]. Further, it was also shown to inhibit FtsZ (filamenting temperature-sensitive mutant Z) protein, thus inhibiting cell division [57,92].

Another isoquinoline alkaloid, ungeremine, isolated from methanolic fractions of *Pancratium Illyricum* L., was found to possess significant antibacterial activity, as it inhibits bacterial topoisomerase, leading to DNA cleavage [92,111].

Piperine, isolated from *Piper nigrum* L. (black pepper) and *Piper longum* L. (Indian long pepper), is a piperidine alkaloid that has demonstrated antimicrobial activity against *Staphylococcus aureus* and synergistically reduced the minimum inhibitory concentration (MIC) values when administered along with fluoroquinolone antibiotics [112]. Its inhibitory effect against MRSA was due to the inhibition of NorA efflux pumps. The synergism of piperine and aminoglycoside antibiotic, namely gentamicin when administered as nano-liposomes, demonstrated high effectiveness against MRSA infection [92,113,114]. Apart from naturally occurring piperine, its synthetic analogues such as 5-(2,2-dimethyl-chroman-6-yl)-4-methyl-penta-2,4-dienoic acid ethyl ester and 5-(2,2-Dimethyl-chroman-6-yl)-4-methyl-2E,4E-pentadienoic acid pyrrolidine were also found to inhibit NorA efflux pumps expressed in *Staphylococcus aureus* [27,115].

Maculine, kokusagine, and dictamine belong to the quinolone class of alkaloids, are primarily found in the stem bark of *Teclea afzelii* Engl., and have demonstrated significant antimicrobial activity. The mode of action of both natural and synthetic quinoline alkaloids involves the inhibition of type II topoisomerase leading to the inhibition of DNA replication [116]. Reserpine, isolated from *Rauvolfia serpentina* (L.) Benth. ex Kurz, is an indole alkaloid, was found to inhibit efflux pumps, and was reported to decrease the fluoroquinolone resistance in *Stenotrophomonas maltophilia*, which was earlier resistant due to over-expression of efflux pumps [117].

The steroidal alkaloids tomatidine and conessine possess antibacterial activities due to potentiating other antibiotics when used in synergism. When used alone or in conjunction with aminoglycoside antibiotics, tomatidine, which is derived from plants belonging to the *Solanaceae* family such as tomato, brinjal, and potato, has demonstrated antimicrobial activity against *Staphylococcus aureus* [118]. It could be used as a potentiator for many antibiotics of different classes, such as ampicillin, cefepime, ciprofloxacin, and gentamicin, when used to treat infections caused by *Pseudomonas aeruginosa*, *Staphylococcus aureus*, or *Enterococcus faecalis* bacteria [92]. Conessine has demonstrated synergistic activity when administered along with antibiotics [92,119]. It has demonstrated resistance-modifying activity against *Acinetobacter baumannii* by inhibiting the *AdeIJK* efflux pump [120].

Sanguinarine is an alkaloid constituent of many plants including *Argemone Mexicana* L., *Chelidonium majus* L., *Macleaya cordata* (Willd.) R.Br., and *Sanguinaria canadensis* L. It has shown antibacterial activity against MRSA strains, and its mechanism involves cell lysis brought about by the release of autolytic enzymes [121]. It was also reported to act as an effective inhibitor of bacterial replication and transcription [122]. Furthermore, Sanguinarine exhibits potent antimycobacterial activities against *Mycobacterium aurum* and *Mycobacterium smegmatis* [123].

Caffeine, a xanthine alkaloid, has shown anti-quorum-sensing activity against *Pseudomonas aeruginosa* by interacting with the quorum-sensing proteins such as LasR and LasI and down-regulating the secretion of its virulence factors [124].

Plant secondary metabolites (PSM) can have additive, antagonistic, or synergistic effects on conventional antibiotics. However, the synergistic effect of PSM with antibiotics is the most preferable interaction in terms of antimicrobial therapies. Two drugs are said to be synergistic when the combined effect they produce is greater than the sum of their individual effects (the phenomenon where the combined effect equals the sum is known as additive effect). Synergistic interaction between two drugs is preferred in the case of antimicrobial therapies, as it allows the use of lower doses of the combination constituents, which not only reduces the duration of antimicrobial therapy but also reduces the chances of dose-dependent toxicity, if any [89]. Many PSMs have been found to have synergistic activities with antibiotics against pathogenic infections. Chanoclavine, an ergot alkaloid, has shown synergistic activity with tetracycline against resistant strains of *Escherichia coli* [125]. Furthermore, 1-4-napthoquinone has demonstrated antimicrobial activity for both Gram-negative and Gram-positive bacteria [126]. It has exhibited synergistic behaviour with carbapenems (imipenem) and cephalosporins (cefotaxime and cefuroxime) against MRSA [13,127].

Alkaloids found in the plant kingdom are structurally very diverse and thus show variability in scale and mode of activity. However, irrespective of the diversification in the mechanism of action, plant alkaloids can be developed into potent antimicrobials, which would not only revive the treatment options but also ensure the development of further resistance is prevented. Table 3 summarizes some of the important plant-derived alkaloids, their target pathogens, and their modes of action.

#### 3.2.2. Phenols

Due to their wide range of pharmacological activities and strong pharmacological effects, plant phenolics are recognised as important bioactive compounds. Plant-derived phenols can be found in simple or polymerized forms and contain an aromatic ring structure with one or more hydroxyl groups. Plant phenolics are categorized into many classes such as simple phenols, phenolic acids, quinones, flavonoids, and tannins. Phenols have proven to be potent against a wide range of diseases such as bacterial infections, cancers, diabetes, and cardiovascular diseases [137,138,139,140]. Plant phenolics have exhibited antimicrobial potency against a variety of microbes by sensitizing them against antibiotics and tuning down the efflux pump activity by acting as potent efflux pump inhibitors.

Simple phenols such as catechol and pyrogallol, which are allelochemicals synthesized by plants, have shown antibacterial activities against three bacterial strains: *Corynebacterium xerosis*, *Pseudomonas putida*, and *Pseudomonas pyocyanea.* Moreover, catechol was found to have an antifungal effect on *Fusarium oxysporum* and *Penicillium italicum* [141]. Furthermore, 4-(4-Hydroxyphenethyl) phen-1,2-diol (2a), a derivative of catechol and pyrogallol, was found to inhibit *Helicobacter pylori* urease enzyme [142]. Resorcinol, isolated from *Ainsliaea bonatii* Beauverd, was found to be effective against MRSA and ESBL *Staphylococcus aureus*. The mode of action was reported to be cell wall disintegration, leading to increased permeability and leakage of intracellular constituents, negatively influencing gene expression and leading to decreased protein synthesis [143]. Resveratrol, a natural phenolic compound, exhibited efflux pump inhibitory activity against various bacterial strains such as CmeABC, a multidrug efflux system of *Campylobacter jejuni*, and efflux pumps of *Mycobacterium smegmatis* [144].

Gallic acid and ferulic acid have been reported to possess significant antimicrobial activities against *Escherichia coli*, *Staphylococcus aureus*, *Listeria monocytogenes*, and *Pseudomonas aeruginosa*, and the mode of action was found to be the disruption of cell membrane via changes in membrane potential [145]. Furthermore, 3-p-trans-coumaroyl-2-hydroxyquinic acid, isolated from *Cedrus deodara* (Roxb. ex D.Don) G.Don, has shown effective antibacterial activity against *Staphylococcus aureus*, and the mechanism of action involves damage to cytoplasmic membrane due to membrane hyperpolarization and loss of membrane integrity, which results in subsequent discharge of intracellular constituents [146]. Chebulinic acid, primarily isolated from *Terminalia chebula* Retz., has been reported to inhibit DNA gyrase of quinolone-resistant *Mycobacterium tuberculosis* [147]. However, the whole study was in silico based, and further insights are needed to unravel its significance as a DNA gyrase inhibitor and anti-tuberculosis agent [92].

Quercetin and apigenin belong to the flavonoid class of plant phenols, which act as antibacterial agents against *Helicobacter pylori* and *Escherichia coli*, and the mechanism of action involves inhibition of d-alanine:d-alanine ligase, an enzyme important for bacterial cell wall assembly [148].

Baicalein is a flavone, primarily isolated from *Scutellaria baicalensis* Georgi, *Scutellaria lateriflora* L., and *Thymus vulgaris* L. It inhibits NorA efflux pumps, thus increasing the efficacy of antibiotics such as β-lactams, ciprofloxacin, and tetracycline against methicillin-resistant *Staphylococcus aureus*. When co-administered with tetracycline, baicalein also shows a synergistic effect against *Escherichia coli* due to inhibition of the efflux pump [103,104].

Biochanin A, an isoflavone, has inhibitory activity against MRSA and has been found to inhibit MRSA efflux pumps by reducing NorA protein expression [149].

Kaempferol, an active flavonoid, has shown potent antimicrobial activity against triazole-resistant *Candida albicans* and MRSA [150,151]. Kaempferol inhibits NorA efflux pump, as does its naturally occurring glycoside derivative, kaempferol rhamnoside, which has a potentiating effect on ciprofloxacin against NorA pumps of *Staphylococcus aureus* [150].

Catechins found in green tea form the basis of the antimicrobial potential of tea extracts. The antimicrobial activity of catechins is attributed to their hydrogen peroxide generation, which ultimately leads to bacterial cell membrane damage [152]. Epigallocatechin gallate (EGCG) is yet another phenolic compound that exhibits antimicrobial activity against MRSA by inhibiting NorA efflux pump [27,92,98]. EGCG has been shown to inhibit DNA gyrase by blocking its β-subunit at the ATP binding site, bacterial efflux pump, and inhibition of chromosomal penicillinases, owing to its multitargeted action against pathogenic microbes [153].

Tannins have been reported to have much more effective antimicrobial action on Gram-positive bacteria than Gram-negative ones. This difference in activity is because of the mode of action of tannins. Tannins pass through the bacterial cell wall and interfere with the metabolism of bacterial cell. On the other hand, double-layered cell walls of Gram-negative bacteria offer much resistance for the tannins to pass through, hence the reduced activity [154]. Curcumin, abundantly found in *Curcuma longa* L., has demonstrated antimicrobial activity against *Escherichia coli* and *Staphylococcus aureus*. The antibacterial activity is attributed to its capacity to damage the membrane by penetrating through the bilayer and increasing the membrane permeability [155].

Phenolics have shown diverse mechanisms against different bacteria ranging from inhibition of efflux pumps, cellular membrane disruption, and inhibition of cell wall synthesis to inhibition of key enzyme biosynthesis. The observed traits of phenolics as antibacterials make them desirable candidates for further in vitro studies. The most significant phenolics with antibacterial activities have been summarized in Table 4.

#### 3.2.3. Organosulfur Compounds

Organosulfur compounds are sulfur-containing organic molecules that are responsible for the strong aromas of *Allium* vegetables such as onions and garlic. They are also present in cruciferous vegetables such as cabbage and broccoli. Several organosulfur compounds such as allicin, ajoene, dialkenyl sulfides, S-allyl cysteine, and isothiocyanates were found to be effective against both Gram-positive as well as Gram-negative bacteria [168,169,170,171]. Investigations have revealed that high-concentration polysulfide-containing plants possess broad-spectrum antibacterial activities [172].

Diallyl thiosulfinate, commonly known as ‘allicin’, is an organosulfur compound that is isolated from *Allium sativum* L. Its antibacterial action has been seen against a variety of pathogenic microbes, including MRSA, *Pseudomonas aeruginosa*, *Streptococcus agalactiae*, *Staphylococcus epidermidis*, and oral pathogens that can cause periodontitis [168,173]. Allicin mainly causes the suppression of sulfhydryl-dependent enzymes, including alcohol dehydrogenase, thioredoxin reductase, and RNA polymerase, which is the primary mechanism of its antibacterial activity. Further, allicin has also been shown to partially inhibit protein and nucleic acid synthesis [174,175].

Ajoene, another organosulfur compound, is not as functionally diverse as allicin. However, it exhibits potency against both Gram-positive as well as Gram-negative bacteria along with some fungal strains, including *Aspergillus niger* and *Candida albicans*. The mechanism of action is the same as that of allicin, as ajoene is also a sulfhydryl-dependent enzyme inhibitor [168].

Isothiocyanates (ITCs) are exclusively abundant in members of the family *Brassicaceae* Burnett. such as broccoli, cabbage, cauliflower, and mustard, and they show activity against oral pathogens as well as *Helicobacter pylori* [170,176,177]. The antimicrobial mechanism of ITCs is not fully understood yet. However, it is speculated that their activity might be due to their reaction with cellular proteins and enzymes, which then hamper the biochemical processes inside the cell. Due to the high electrophilicity of an ITC carbon atom, it can react with amines, thiols, and hydroxyl groups of cellular proteins [170]. Table 5 lists some of the important organosulfur compounds that have been found to have antimicrobial activities against different pathogenic microbes.

#### 3.2.4. Terpenes

Terpenes are aromatic compounds found in many plants and are responsible for the characteristic smell of many plants, such as cannabis, pine, and lavender, as well as fresh orange peel. Terpenes are commonly distributed in nature, in nearly all living forms, and perform a variety of functions in cells. Apart from being primary structural components of cells (cholesterol and steroids in cellular membranes), they also act as functional molecules such as carotenoids, quinones, and retinal in photosynthesis, electron transport, and vision, respectively [183].

Normally, terpenes have demonstrated more potent activity for Gram-positive than Gram-negative bacteria and bring about their antibacterial effects mainly via lipophilic features. Monoterpenes change membrane structure by changing their composition, which increases fluidity and permeability and causes changes in the topology of membrane proteins, causing disruptions throughout the respiratory chain [184]. Carvacrol is commonly found in the essential oils of *Thymus vulgaris* L., *Lepidium flavum* Torr., *Citrus aurantium* (Spreng.), Balle ssp. *Bergamia*, and *Origanum vulgare* L., among other plants. It has demonstrated antibiofilm development activity against *Staphylococcus aureus* and *Salmonella typhimurium* and is reported to have activity against tobacco mosaic virus and cucumber mosaic virus [185,186]. Carvacrol has also been shown to be effective against food-borne pathogens such as *Escherichia coli*, *Salmonella*, and *Bacillus cereus* [124].

Thymol, found as an essential oil component of *Thymus vulgaris* L, has shown antibacterial effects on tetracycline-resistant *Salmonella typhimurium* and *Escherichia coli*, penicillin-resistant *Staphylococcus aureus*, and erythromycin-resistant *Streptococcus pyogenes*. The mechanism of action, as per many studies, involves disintegration of cell membranes [187,188].

Ursolic acid, a pentacyclic triterpene, possess broad-spectrum antibacterial activity. It was shown that ursolic acid has disorganising effects on *Escherichia coli* membrane [189]. Eugenol and cinnamaldehyde are yet more important terpenes present in plant essential oils and have shown activity against a wide range of pathogens including *Helicobacter pylori*, causing damage to the cell membrane [190,191]. Eugenol has been shown to inhibit biofilm formation by MRSA and MSSA clinical strains as well as the synthesis of virulence factors by *Pseudomonas aeruginosa* [190,192]. The mechanism of eugenol action involves damage to bacterial membrane, followed by leakage of cellular contents. As for cinnamaldehyde, the compound works by damaging the membrane, decreasing the membrane potential, and alterations in metabolic activity [193]. Some of the most significant terpenes having antibacterial effects are listed in Table 6.

## 4. Himalayan Medicinal Plants as a Reservoir of Phytochemicals for Novel Antimicrobial Drug Discovery

### 4.1. Plant Diversity of Indian Himalayas

The Indian Himalayas are one of the thirty-six designated biodiversity hotspots globally [16]. Spread over an area of 3000 km from Northern Pakistan to North East India, the region spans incredible variations in climate across its course. Geographically, the entire mountain range has been divided into two regions: the Eastern Himalayas, which span from Nepal, Tibet, Bhutan, West Bengal, Assam, and Arunachal Pradesh to Northern Myanmar; and the Western Himalayas, which include parts of Uttarakhand, Northwest Kashmir, and Northern Pakistan [16].

The Indian Himalayan region is home to an estimated 10,000 species of vascular plants, out of which 3160, accounting for almost 1/3^rd^ of the total plant species found, are endemic to the region [197]. Additionally, 71 genera and 5 plant families are also endemic to the area. The endemic plant families include *Trochodendraceae* Eichler, *Hamamelidaceae* R. Br., *Butomaceae* Mirb., and *Stachyuraceae* J. Agardh. The largest family of flower-bearing plants in the region is *Orchidaceae* Juss., with an estimated number of 750 species [197]. Among the 5725 species of angiosperms endemically found in India, 3471 species are hosted by the Himalayas themselves. Moreover, among the 147 genera of angiosperms that are endemic to India, 71 are found exclusively in the Himalayan region [198]. The Himalayas host all the conifer (gymnosperms) flora of India except for *Podocarpus wallichianus* C.Presl and *Podocarpus neriifolius* D.Don, which are found in peninsular India and the Andamans, respectively. Among the gymnosperm shrubs, *Ephedra gerardiana* Wall. ex Klotzsch & Garcke is exclusively distributed in the Himalayas and is highly revered as a medicinal plant due to its alkaloid ephedrine [199]. Among the pteridophytes, the Eastern Himalayas contain about 847 taxa in 179 genera, followed by the Western Himalayas, which contain 340 taxa in 101 genera of pteridophytes [200]. Of the 2000 species of mosses (bryophyte) found in India, the Eastern Himalayas contain 1030 species and 751 species are distributed in the Western Himalayas [201]. About 30% of the total liverwort population is maximally distributed in the Eastern Himalayas followed by the Western Himalayas and the Western Ghats [202].

Around 30% of India’s land area, including biodiversity hotspots such as the Himalayas, the Western Ghats, and the Nicobar Islands, is still unexplored and unrecorded in terms of its floral richness. Therefore, our understanding of the delicate ecosystems of these hotspots is still insufficient [16].

### 4.2. Medicinal Plant Resources of Himalayas and Alternate Systems of Medicine

The Indian subcontinent possesses one of the oldest and most well-structured medical systems, which originated more than 5000 years ago [203]. The vast information on medicine is backed by different traditional medicinal practices such as Ayurveda and Unani and various literary manuscripts such as *Charak Samhita*, *Sushruta Samhita*, *Dhanvantri*, and *Nighatu* [204,205]. These scriptures provide a solid foundation for traditional medicinal practices in India [206]. Various communities in India, both tribal and urban, rely on traditional medicine, and it has long been an important element in the treatment of diseases and disorders. Around 25000 phytocompounds are used as herbal formulations in rural Indian traditional medicine, particularly in tribal populations [207]. Of these phytocompounds, only 5–10% have been confirmed scientifically [208]. Due to the rising interest in adopting traditional medicine globally, government institutions in India have made attempts to validate the therapeutic efficiency of the drugs used in traditional medicine [209]. The Himalayan region is home to many endemic human populations, and due to the remoteness of the area, the people have been relying on forest products for multiple needs, including the ethnomedicinal use of plants for disease treatment, as a result of which the people of the Himalayas have a strong belief in traditional herbal medicine [210,211].

The Indian Himalayas foster around 10,000 species of higher plants, of which 1748 species reportedly have medicinal properties [18,19]. Medicinal plants of the region have played fundamental roles in the disease treatment of the people living in and around the Himalayan mountain range [19]. The vegetation of the area is determined by the climate and weather conditions of the area. For instance, the North-Western Himalayas, including the areas of Ladakh and Gilgit, have weather conditions ranging from mild summers to severely cold winters, and the medicinal flora are represented by *Achillea millefolium* L., *Bunium persicum* (Boiss.) B. Fedtsch., *Picrorhiza kurroa* Royle ex Benth., *Juniperus communis* L., and *Ephedra gerardiana* Wall. ex Klotzsch & Garcke [212]. The Western Himalayan region, including Jammu and Kashmir, Himachal Pradesh, Garhwal, and Kumaon Himalaya, experiences warm humid summers and cold humid winters, and the medicinal flora are primarily represented by *Saussurea costus* (Falc.) Lipsch., *Colchucum luteum* Baker, *Atropa acuminata* Royle ex Lindl., and *Physochlaina praealta* (Decne.) Miers. On the other hand, the Eastern Himalayas, comprising areas such as Darjeeling, parts of Assam, Sikkim, and Arunachal Pradesh, are characterized by warm summer and cool winter. Hence, the vegetation is represented predominantly by *Aquilaria malaccensis* Benth., *Coptis teeta* Wall., and *Panax pseudoginseng* Wall. [16]. In the adjoining Himalayan region of north-western Pakistan, medicinal plants such as *Berberis lyceum* Royle, *Achillea millefolium* L., *Bergenia ciliata* (Royle) A.Braun ex Engl., and *Aloe vera* L. have been reported to be used against urinary tract infections due to their antimicrobial activity against *Staphylococcus aureus* and *Escherichia coli* [140]. Further, medicinal plants such as *Impatiens glandulifera* Royle, *Artemisia scoparia* Waldst. & Kit., *Ageratum conyzoides* L., and *Achillea millefolium* L. have been reported to be used as treatment options for various ailments such as urinary tract infections, cardiac diseases, baldness, abortion and miscarriage jaundice, hepatitis, typhoid, fever, and tuberculosis [211].

In India, around 17,000 species of higher plants have been discovered, of which 7500 plant species have been found to have medicinal properties, which is the highest total-plants-to-medicinal-plants proportion so far reported [16,17]. The maximum population of medicinal plants (1717 species) has been reported at an elevation of 1800 m. Traditional medical practices of the Indian subcontinent use many medicinal plants, and Ayurveda alone has reported 2000 medicinal plant species. One of the oldest written documents on herbal medicine, the Charak Samhita, documents 340 herbal drug productions and their aboriginal uses [213]. The rich diversity of medicinal plants in the Himalayas gave rise to the traditional medicine practices such as Ayurveda and Unani. Apart from the widely followed systems of traditional medicine, various local systems of practices based on the cultural demography have also developed. For instance, the traditional healers of Ladakh region (North-Western Himalayas) are known as “amchies”, those who practice in Kashmir Valley are known as “hakeems”, and those in Jammu are called “veds”. These traditional practices came into existence primarily because of the absence of modern medicine in past times and are still carried forward to this date as a part of tradition [214].

## 5. Antimicrobial Profile of Himalayan Medicinal Plants

One of the main causes of clinical mortality in humans has been infectious diseases. Moreover, with the emergence of multidrug-resistant microbes, the existing antimicrobial therapies have been rendered inactive, which has made the development of new antimicrobials necessary [26]. In the pursuit of novel antimicrobials, plants blessed with a plethora of secondary metabolites offer a vast array of phytochemicals to be screened for novel antimicrobials and developed into new antimicrobial therapies [102]. Humans have been using plants for remedial measures against various ailments for generations, as a result of which many forms of traditional medicines came into existence. These herbal medicines constitute a major part of traditional medical practices [206,211]. The Indian Himalayan region comprises 31% native, 15.5% endemic, and 14% threatened plant species [204]. The floristically rich Himalayan region is a potential source of many drug-yielding plants [215]. Many of the medicinal plants in the Himalayas have shown potent antimicrobial activity against pathogenic microbes [14,216].

Angiosperms such as *Acorus calamus* L. (asarone), *Aegle marmelos* (L.) Corrêa (rutacin), *Arnebia euchroma* (Royle ex Benth.) I.M.Johnst. (shikonin), *Berberis* L. sp. (berberine), *Callicarpa macrophylla* Vahl (sesquiterpenes and triterpenes), *Curcuma caesia* Roxb. (cinnamate), *Hedychium spicatum* G.Lodd. (limonene, linalool), *Inula racemosa* Hook.f. (isoalantolactone), *Jasminum officinale* L. (jasminol, lupeol), *Myrsine semiserrata* Wall. (embelic acid), *Nardostachys jatamansi* (D.Don) DC. (jatamansic acid), and *Piper longum* L. (piperine) are a few of the candidate phytochemicals that have shown potent antimicrobial activities. *Prunus cornuta* (Wall. ex Royle) Steud. and *Quercus semecarpifolia* Sm. have shown antibacterial activity against *Acinetobacter baumannii*, *Salmonella enterica*, and *Escherichia coli* [212].

Gymnosperm plants such as the species of *Cycas* L. and *Ginkgo* L., *Sabina chinensis* (L.) Antoine, *Cedrus deodara* (Roxb. ex D.Don) G.Don, *Pinus bungeana* Zucc. ex Endl., *Platycladus orientalis* (L.) Franco, and *Torreya grandis* Fortune ex Lindl. have shown antimicrobial activities. The essential oil ‘turpentine’ obtained from plants such as *Abies balsamea* (L.) Mill., *Pinus brutia* Ten., and *Pinus roxburghii* Sarg. has demonstrated antimicrobial activity against MRSA [199].

Among the pteridophytes, *Adiantum philippense* L., *Adiantum caudatum* L., *Adiantum incisum* C. Presl., and *Adiantum venustum* D.Don have shown strong antimicrobial activity against pathogens, causing food-borne infections [217]. Members of the genus *Dryopteris* have shown activity against *Pseudomonas aeruginosa* [218]. *Equisetum arvense* L. has shown activity against *Escherichia coli*, *Staphylococcus aureus*, *Klebsiella pneumoniae*, *Pseudomonas aeruginosa*, *Salmonella enteritidis*, *Aspergillus niger*, and *Candida albicans* [219].

Many bryophytes have been used traditionally for inflammation, heart disease, digestive problems, lung, and skin diseases [220]. However, some bryophytes (mosses) have shown antimicrobial properties [221]. *Marchatia polymorpha* L. has demonstrated antimicrobial activity against *Escherichia coli*, *Staphylococcus aureus*, *Proteus mirabilis*, *Aspergillus niger*, *Aspergillus flavus*, and *Candida albicans* [222]. Some antimicrobial bioactive compounds such as polygodial, norpiguisone, and lunularin have been isolated from *Porella platyphylloidea* (L.) Pfeiff., *Conocephalum conicum* (L.) Dumort, and *Lunularia cruciate* (L.) Dumort. ex Lindb.

Medicinal plants are still being used in domestic households for many infectious diseases. For instance, paste of *Rheum emodi* Wall. is used to cure abscesses and boils in many parts of the North-Western Himalayas, particularly in Kashmir Valley; a fermented product of *Viola odorata* L. is used to treat respiratory tract infections; and roots of *Juglans regia* L. are used to treat gum infections [19]. Despite the availability of modern antibiotics, many parts of the Himalayan region, particularly the tribal population, still practice and prefer herbal medicine over modern antibiotics. Although many plant species of Himalayan medicinal plants have been investigated for their antimicrobial activities, given the medicinal plant diversity of the Himalayas, extensive research is needed to explore the untapped reserve of phytochemicals produced by the medicinal plants. The phytochemicals could act as novel antimicrobials, antibiotic potentiators, or resistance breakers. Table 7 summarizes selected medicinal plants of the Indian Himalayas that have shown potency as novel antimicrobials.

## 6. Challenges of Using Phytochemicals as Medicine

### 6.1. Effects of Climate Change

The productivity of medicinal plants is susceptible to changes in climatic conditions, leading to varied responses among different species. This could result in decreased biomass production along with changes in the production of secondary metabolites, thus impacting the quality and safety of herbal medicinal products [340]. Abiotic stresses such as extreme temperatures, increased CO_2_ concentration, and drought conditions influence secondary metabolite production [341]. Since the production of phytochemicals greatly depends on the physiological condition of a plant, the plant response to any of these stress conditions would therefore result in the increase or decrease in the production of phytochemicals from its normal value and hence the compromised efficacy and safety of the prepared herbal formulation [342]. Climate change can severely affect the composition as well as the production of secondary metabolites by the plant, which in the long term may compromise or altogether abrogate the medicinal value of the plant. Further, climate change along with unsustainable harvesting practices can drive plant species to extinction. Therefore, in order to minimize the long-term effects of climate change, conservation efforts, sustainable harvesting practices, preservation of traditional knowledge, and climate change mitigation measures are highly recommended [340].

### 6.2. Toxicity of Herbal Medicine

While herbal medicines are generally considered safer than synthetic counterparts, they can still pose cellular toxicity risks and cause adverse effects on the human system. Toxicity issues may arise from the specific effects of certain phytochemicals in addition to the issues of self-medication, unqualified practitioners, sub-standard products, and improper dosages [343]. Different toxicity issues have been reported with herbal formulations, such as nephrotoxicity induced by a diterpenoid epoxide produced by *Tripterygium wilfordii* Hook., renal calcification caused by *Guaiacum officinale* L., and *Arctostaphylos uva-ursi* L. [344]. Neurotoxicity has been observed with plant species such as *Catharanthus roseus* (L.) G. Don, *Papaver somniferum* L., and *Cannabis indica* Mure [345,346]. Cardiotoxicity has been reported with plants such as *Ephedra distachya* L., *Mandragora officinarum* L., and *Aconitum napellus* L., [347]. Similarly, hepatotoxicity has been associated with *Teucrium chamaedrys* L., *Scutellaria baicalensis* Georgi, and *Larrea tridentata* (DC.) [348].

Apart from inherent toxicity issues, related contamination could occur during product development, which includes heavy metal and microbial contamination, plant misidentification, and economically motivated adulteration [349]. Therefore, to assess the safety of herbal formulations, different techniques could be employed such as predictive toxicology approaches involving in silico modelling. Omics approaches for toxicology studies including toxicogenomics, toxicometabolomics, and toxicoproteomics could be used for early prediction of toxicity in product development [349].

### 6.3. Other Challenges and Regulation

The introduction of phytochemicals in modern medicine poses a number of challenges in terms of standardizing plant extracts to ensure the consistent quality of the end product. The identification and isolation of bioactive compounds, followed by the elucidation of their mechanisms of action, is yet another challenging task due to the presence of diverse compounds in them. Different phytochemicals can interact to work in synergy to bring about their antimicrobial effects, an understanding of which will be crucial in the development of combination therapies. Insufficient scientific evidence and intellectual property protection pose additional hindrances to the integration of herbal formulations in modern medicine. Most of these challenges could be overcome by creating proper checks and balances in the form of comprehensive regulatory bodies, quality control measures, conduction of clinical trials, intellectual property protection, and integration of traditional knowledge with scientific advancements [350].

To set forth a proper safety assessment plan and regulatory laws for manufacturing herbal medicine, different regulatory bodies around the globe such as the International Life Sciences Institute, Washington, DC, USA; International Union of Pure and Applied Chemistry, North Carolina, USA; European Medicines Agency, Amsterdam, The Netherlands; and European Food Safety Authority, Parma, Italy, have issued guidance documents for the assessment of safety of herbal medicine [351]. In the USA, the sale and purchase of herbal medicines is regulated by the Dietary Supplement Health and Education Act of 1994 [352]. In the European Union, the production and marketing of herbal drugs are regulated by various national regulatory bodies such as the Committee on Herbal Medicinal Products (HMPC), which is a part of the European Medicines Agency [349]. Similarly, Canada has Natural Health Products Regulations (NHPR), and Australia has the Therapeutic Goods Administration (TGA) as the regulatory authority to assess and ensure the manufacturing and marketing of herbal drugs [353,354]. In India, the Ministry of AYUSH is the regulatory authority, which provides licenses for the manufacturing and marketing of herbal drugs [355].

## 7. Methodology

In the present study, three databases, namely PubMed, DOAJ, and Google Scholar, were searched by using specific keywords: “Antimicrobial resistance”, “plant antimicrobials”, and “Himalayan medicinal plants”. Collectively, a total number of 4878 articles, both reviews as well as original research articles, published from the year 1940 to 2023, were identified. The selection of articles was rigorously conducted as per the focus of the review article, and only those articles published in peer-reviewed journals were included in the study to ensure the quality of the work.

Furthermore, the chemical structures and formulae of phytocompounds were sourced from PubChem by using their common as well as IUPAC names (wherever necessary), and their respective PubChem IDs were assigned to the compounds in the tabular form. The botanical names of the plants mentioned in the study have been cross-verified with the International Plant Names Index (IPNI).

## 8. Conclusions and Future Prospects

Since the advent of antibiotics, the development of antibiotic resistance and subsequent pursuit of novel antimicrobials has always been a run-and-chase scenario. The discovery of every antibiotic was followed by its resistance development in bacteria. To overcome the resistance against antimicrobials, several anti-resistance measures were taken up; one of the approaches was to develop inhibitors. Clavulanic acid, produced by *Streptomyces clavuligerus*, though far less effective as an antimicrobial agent by itself, proved to be very efficient in augmenting the activity of antibiotics in combination therapies against bacteria producing serine β-lactamases. However, the inefficiency of clavulanic acid in inhibiting metallo-β-lactamases made it a weak option for overcoming the multifaceted mechanisms of antimicrobial resistance. Plant secondary metabolites, on the other hand, are structurally and functionally dynamic entities that have shown broad-spectrum antimicrobial activities.

This study summarizes the timeline of antibiotic discovery and the subsequent emergence of resistance among microbes, along with the mechanistic details of various resistance determinants expressed by multidrug-resistant pathogens. This study further shows the need to explore plants as alternative sources of antimicrobials and sums up the mechanistic effects of various phytochemicals on microbial strains, including those expressing resistance phenotypes. Plant phytochemicals have shown multifaceted effects on microbial cells, such as alkaloids imposing their antimicrobial effects by inhibiting efflux pumps, nucleic acid synthesis, enzyme activities, ATP synthesis, cell-to-cell communication, cell wall biosynthesis, and jeopardizing the cell division machinery. Phenols act as antimicrobials by inhibiting metabolically important enzymes, efflux pumps, and cell wall biosynthesis; inactivating enzymes such as DNA gyrase and penicillinases; and increasing the membrane permeability, eventually leading to cell death. Organosulfur compounds inhibit nucleic acid synthesis and act as potent enzyme inhibitors, while terpenes mostly exert their action by compromising the membrane integrity of microbial cells.

Plant-based antimicrobials have attracted significant attention due to the reduced potency and increasing toxicity of synthetic antimicrobials. Plant-based antimicrobial formulations have emerged as a boon in medical sciences, as they are easily available and have almost no side effects. The limited target specificity of existing synthetic antibiotics can be overcome by the broad-spectrum antibacterial action of phytochemicals. The poor adaptability of bacteria, fungi, and viruses to a plant-based antimicrobial regime might be the reason for the efficacy of herbal treatment against diseases. Therefore, synergistic combinations of synthetic antimicrobials and chemically defined phytochemicals will not only help to deal with global antimicrobial resistance but will also assure no further resistance development. This study provides a comprehensive overview of the medicinal plants of the Indian Himalayan region. The region, as discussed, possesses a mammoth repository of medicinal plants. The Indian Himalayan region happens to be one of the world’s 36 biodiversity hotspots. With such boundless plant resources available, the tribal communities in particular have developed traditional ways to use these plants for medicinal purposes. However, detailed studies deciphering the medicinal importance of these plants in the region have been scarce. In this study, 123 plants, native to the Himalayan region, with antimicrobial properties have been described along with their reported bioactive compounds and the target pathogens against which they are active. However, details on their mechanistic roles as antimicrobials are not available in every case where the effects of their crude extracts have been described. Further research is required on the identification of active substances and the underlying mechanisms of action, and efficacy analysis during in vivo applications is highly necessary to assist the pursuance of potent antimicrobials. This not only will allow the development of novel plant-based antimicrobials but also opens up the possibilities for developing combination therapies as multitarget solutions, which will greatly help to combat antimicrobial resistance in bacterial strains.

## Figures and Tables

**Figure 1 pharmaceuticals-16-00881-f001:**
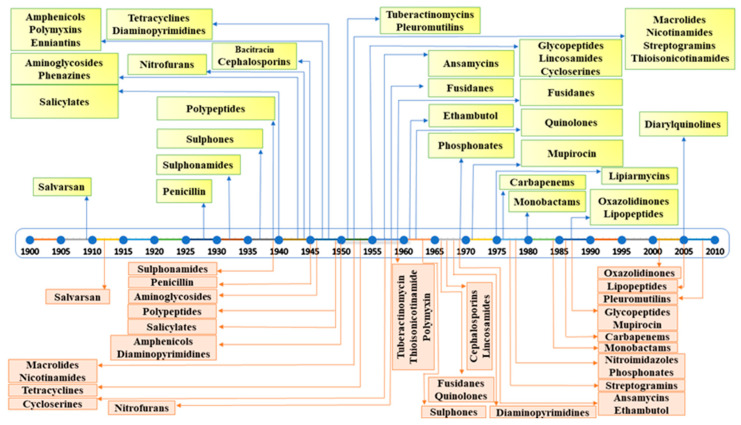
Timeline of introduction of antibiotics (*shown in yellow*) and the emergence of antimicrobial resistance (*shown in light brown*).

**Figure 2 pharmaceuticals-16-00881-f002:**
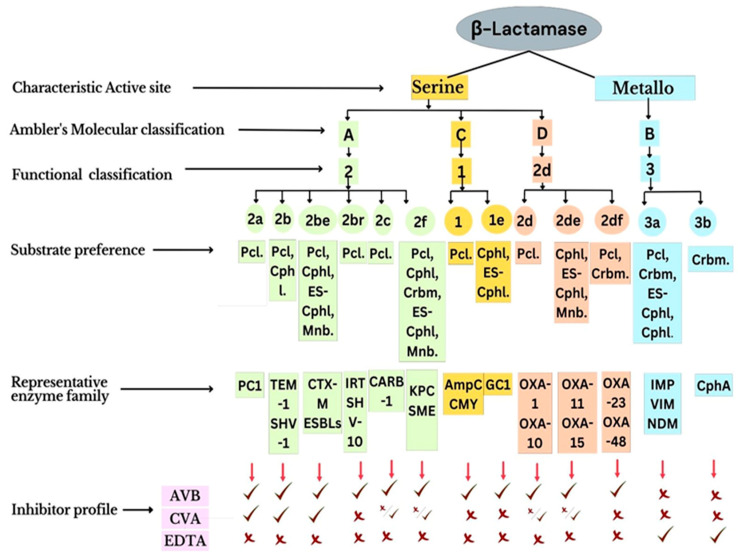
Molecular and functional classification of β-lactamases. Adapted with permission from [60]. [Pcl—Penicillin, Cphl—Cephalosporin, ES-Cphl—Extended-spectrum cephalosporins, Mnb—Monobactams, Crbm—Carbapenem, AVB—Avibactam, CVA—Clavulanic acid, EDTA—Ethylenediamine tetra acetic acid].

**Figure 3 pharmaceuticals-16-00881-f003:**
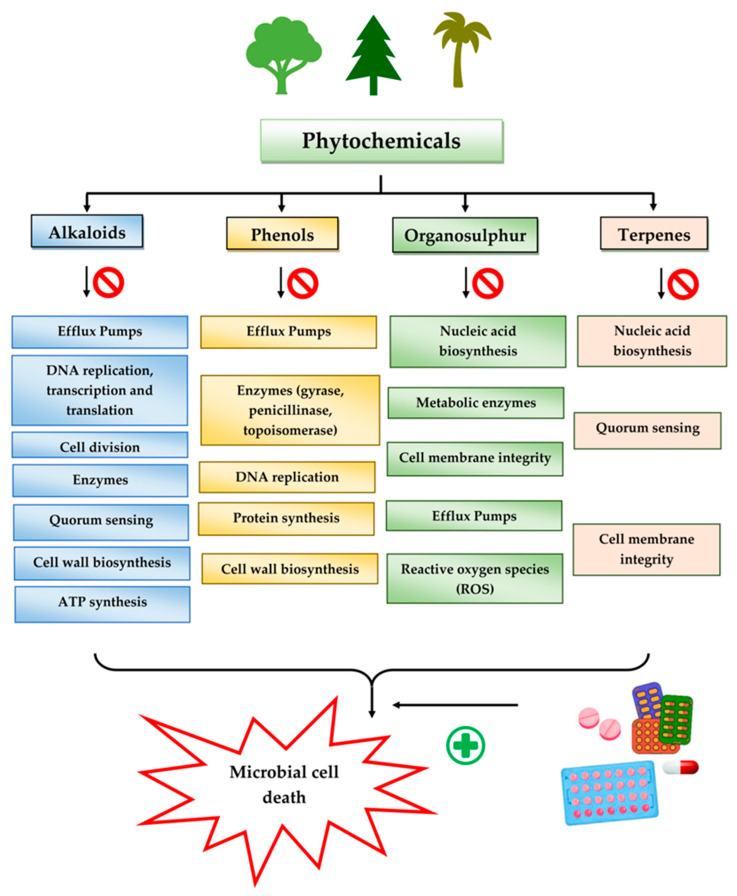
Flowchart of plant-derived phytochemicals and their mechanistic role as antimicrobials.

**Table 1 pharmaceuticals-16-00881-t001:** List of efflux pump families and representative candidates from selected candidates of *ESKAPE* pathogens.

Pathogenic Bacteria	Efflux Pump Family	Representative of Efflux Pump	Antibiotic Effluxed	References
*Enterococcus faecium*	ABC	*EfrAB*	Acriflavine, ciprofloxacin, daunomycin, doxorubicin, doxycycline, norfloxacin, tetraphenylphosphonium	[33]
*Staphylococcus aureus*	ABC	*Isa(E)*	Linosamide, pleuromutilin, streptogramin A	[34]
		*Msr(A)*	Macrolides, telithromycin	[34]
	MATE	*MepA*	Biocides, ethidium bromide, fluoroquinolones	[35]
	MFS	*NorA*	Fluoroquinolones	[36,37]
		*QacA*	Acriflavine, chlorhexidine, ethidium bromide, quaternary ammonium compounds	[37]
*Klebsiella pneumoniae*	MATE	*KetM*	4, 6 Diamidino- 2- phenyl indole	[38]
	MFS	*KpnGH*	Ceftazidime, cefepime, streptomycin, tetracycline	[39]
	RND	*OqxAB*	Chloramphenicol, fluoroquinolones	[40]
	SMR	*KnpEF*	Benzalkonium chloride, cefepime, chlorhexidine, erythromycin, streptomycin, tetracycline, triclosan	[39]
*Acinetobacter baumannii*	ABC	*MacAB- TolC*	Macrolides	[41]
	MATE	*AbeM*	Acriflavine, aminoglycosides, daunomycin, doxorubicin, fluoroquinolone	[42]
	MFS	*CraA*	Chloramphenicol	[43]
	RND	*AdeABC*, *AdeFGH*, *AdeIJK*	Aminoglycosides, beta-lactams, fluoroquinolones, macrolides, tetracycline, biocides	[44,45]
*Pseudomonas aeruginosa*	RND	*MexAB- OprM*,	Aminoglycosides, beta-lactams, chloramphenicol, fluoroquinolones, macrolides, sulfonamides, tetracyclines, tigecycline	[46]
		*MexXY- OprM/A*, *MexCD- OprJ*, *MexEF- OprN*	Trimethoprim, biocides, ethidium bromide	[47]
*Escherichia coli*	ABC	*MacAB- TolC*	Macrolides	[48]
	MFS	*MdfA*	Chloramphenicol, doxorubicin, norfloxacin, tetracycline	[49]
		*QepA/QepA2*	Fluoroquinolones	[50]
	RND	*AcrAB- TolC*	β-lactams, chloramphenicol, fluoroquinolones, macrolides, novobiocin, tetracycline, tigecycline	[51,52]
		*OqxAB*	Chloramphenicol, fluoroquinolones	[40]
	SMR	*EmrE*	Acriflavine, ethidium bromide, quaternary ammonium compounds	[53,54]

**Table 2 pharmaceuticals-16-00881-t002:** List of enzymes and their classes involved in the modification and inactivation of antibiotics.

Enzyme Class	Type	Substrate Antibiotic Class	Representative
Hydrolases	β-lactamases	β-lactam	Penicillin
			Cephalosporin
			Carbapenem
	Macrolide esterases	Macrolide	Erythromycin
			Roxithromycin
			Azithromycin
	Epoxidases	Epoxide	Fosfomycin
Transferases	Acetyltransferases	Aminoglycoside	Gentamicin
			Kanamycin
			Amikacin
		Chloramphenicol	Chloramphenicol
		Streptogramin	Group A streptogramins
	Phosphotransferases	Aminoglycoside	Gentamicin
			Kanamycin
			Amikacin
		Macrolide	Erythromycin
			Roxithromycin
			Azithromycin
		Rifamycin	Rifampin
			Rifabutin
			Rifapentine
		Peptide	Colistin
			Polymixin B
	Thiol S-transferases	Epoxide	Fosfomycin
	Nucleotidyltransferases	Aminoglycoside	Gentamicin
			Kanamycin
			Amikacin
		Lincosamide	Lincomycin
			Clindamycin
			Pirlimycin
	ADP-ribosyltransferases	Rifamycin	Rifampin
			Rifabutin
			Rifapentine
	Glycosyltransferases	Macrolide	Erythromycin
			Roxithromycin
			Azithromycin
		Rifamycin	Rifampin
			Rifabutin
			Rifapentine
Redox enzymes	Monooxygenases	Tetracycline	Tetracycline
			Oxytetracycline
			Doxycycline
		Rifamycin	Rifampin
			Rifabutin
			Rifapentine
		Streptogramin	Group A streptogramins
Lyases	Lyases (Virginiamycin B lyase)	Streptogramin	Group B streptogramins

**Table 3 pharmaceuticals-16-00881-t003:** List of important alkaloid classes of plant-based antimicrobial agents reported and their sources, pathogenic targets, and mechanisms of action.

Bioactive Compound	Chemical Formula	PubChem CID	Chemical Structure *	Plant Source	Target Pathogen	Mode of Action	References
Conessine	C_24_H_40_N_2_	441082	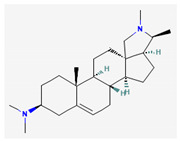	*Holarrhena antidysenterica* (G. Don) Wall. ex A. DC.,*Holarrhena floribunda* (G.Don) T. Durand & Schinz, *Holarrhena pubescens* Wall. ex G. Don,*Funtumia elastica* (Preuss) Stapf.	*Pseudomonas aeruginosa*	Efflux pump inhibitor	[120,128]
Piperine	C_17_H_19_NO_3_	638024	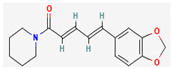	*Piper sylvaticum* Roxb.	Methicillin- resistant *Staphylococcus aureus* (MRSA)	Efflux pump inhibitor	[92,114]
Berberine	C_20_H_18_NO_4_^+^	2353	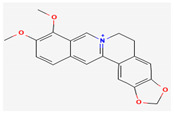	*Berberis lycium* Royle	*Escherichia coli*	Cell division inhibitor, protein, and DNA synthesis inhibitor	[129,130]
Lysergol	C_16_H_18_N_2_O	14987	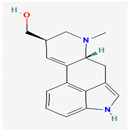	*Convolvulaceae* Juss.	*Escherichia coli*	Efflux pump inhibitor	[125]
8-epidiosbulbin E-acetate	C_21_H_24_O_7_	131751666	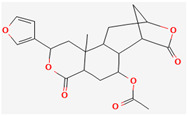	*Dioscorea bulbifera* L.	*Escherichia coli*, *Enterococcus faecalis*, *Pseudomonas aeruginosa*, *Shigella.*	Plasmid curing(R-plasmids in *Escherichia coli* and *Enterococcus faecalis*)	[57,107]
Reserpine	C_33_H_40_N_2_O_9_	5770	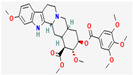	*Rauvolfia serpentina* (L.) Benth. ex Kurz	*Staphylococcus* sp., *Streptococcus* sp.	Efflux pump inhibitor	[131]
Tomatidine	C_27_H_45_NO_2_	65576	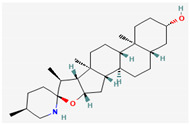	*Solanum* L. sp.	*Listeria*, *Bacillus* and *Staphylococcus* sp.	ATP synthase inhibitor	[118,132]
Dictamnine	C_12_H_9_NO_2_	68085	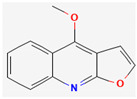	*Teclea afzelii* (Engl.)	*Escherichia coli*, *Microsporum audorium*, *Bacillus subtilis*, *Mycobacterium smegmatis*	Inhibition of Type II topoisomerase enzyme and inhibition of DNA replication	[92]
Kokusagine	C_13_H_9_NO_4_	5318829	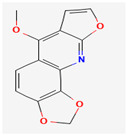	*Teclea afzelii* (Engl.)
Maculine	C_13_H_9_NO_4_	68232	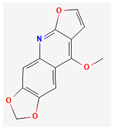	*Teclea afzelii* (Engl.)
Sanguinarine	C_20_H_14_NO_4_^+^	5154	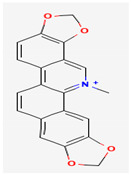	*Chelidonium majus* L., *Sanguinaria canadensis* L., Macleaya *cordata* (Willd.) R. Br.	MRSA, *Mycobacterium aurum* and *Mycobacterium smegmatis*	Compromising cytoplasmic membrane, cell lysis, replication, and transcription inhibition	[92]
Chanoclavine	C_16_H_20_N_2_O	5281381	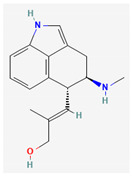	*Ipomoea muricata* (L.) Jacq.	MDR *Escherichia coli*	Efflux pump inhibition	[92]
Caffeine	C_8_H_10_N_4_O_2_	2519	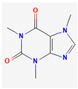	*Camellia sinensis* (L.) Kuntze	*Pseudomonas aeruginosa*	Inhibition of quorum-sensing proteins *LasR* and *LasI* andinhibition of bacterial virulence factors	[124]
Caranine	C_16_H_17_NO_3_	441589	* 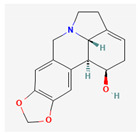 *	*Clivia miniata* (Lindl.) Verschaff.,*Crinum bulbispermum* (Burm.f.) Milne-Redh. & Schweick.	*Candida dubliniensis*	NA	[133]
Evodiamine	C_19_H_17_N_3_O	442088	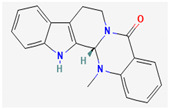	*Evodia aromatica* (Sonn.) Pers.	*Streptococcus pneumoniae*	Inhibition of ATP-dependent MurE ligase of *Mycobacterium tuberculosis*, an enzyme required for the biosynthesis of peptidoglycan	[134]
Chanoclavine	C_16_H_20_N_2_O	5281381	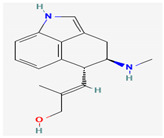	*Ipomoea muricata* (L.) Jacq.	MDR *Escherichia coli*	Efflux pump inhibition	[92]
Evocarpine	C_23_H_33_NO	5317303	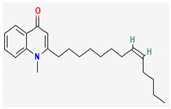	*Evodia aromatica* (Sonn.) Pers.	*Streptococcus pneumoniae*	Inhibition of ATP-dependent MurE ligase of *Mycobacterium tuberculosis*, an enzyme required for the biosynthesis of peptidoglycan	[134]
Voacafricines A & B				Fruits of *Voacanga Africana* Stapf.	*Staphylococcus aureus*	NA	[133]
Thalicfoetine				Roots of *Thalictrum foetidum* L.	*Bacillus subtilis*	NA	[135,136]

* Chemical structures of compounds have been taken from PubChem; www.pubchem.ncbi.nlm.nih.gov (Accessed on 10 December 2022).

**Table 4 pharmaceuticals-16-00881-t004:** List of important phenolic classes of plant-based antimicrobial agents reported and their sources, pathogenic targets, and mechanisms of action.

Bioactive Compound	Chemical Formula	PubChem CID	Chemical Structure *	Plant Source	Target Pathogen	Mode of Action	References
Myricetin	C_15_H_10_O_8_	5281672	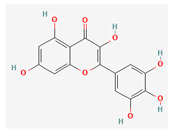	*Myricaceae* Rich. ex Kunth., *Anacardiaceae* R.Br., *Polygonaceae* Juss., *Pinaceae* Spreng. ex F.Rudolphi., *Primulaceae* Batsch ex Borkh.	*Mycobacterium smegmatis*	Efflux pump inhibitor	[144]
Baicalein	C_15_H_10_O_5_	5281605	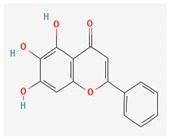	*Thymus vulgaris* L., *Scutellaria baicalensis* Georgi, *Scutellaria lateriflora* L.	Methicillin-resistant *Staphylococcus aureus*	Inhibition of the *NorA* efflux Pump	[103,104]
4′,6′-Dihydroxy-3′,5′-dimethyl-2′-methoxychalcone	C_18_H_18_O_4_	10424762	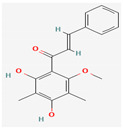	*Dalea versicolor* Zucc.	*Staphylococcus aureus,* *Bacillus cereus.*	Inhibition of *NorA* efflux pump	[92,156]
Epigallocatechin gallate	C_22_H_18_O_11_	65064	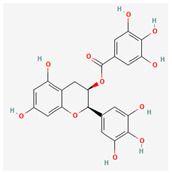	*Camellia sinensis* (L.) Kuntze	Methicillin-resistant *Staphylococcus aureus*	Inhibition of *NorA* efflux pump, inhibition of chromosomal penicillinase and DNA gyrase	[98,153]
Chebulinic acid	C_41_H_32_O_27_	72284	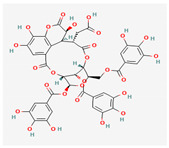	*Terminalia chebula* Retz.	Quinolone-resistant mutants of *Mycobacterium tuberculosis*	Inhibition of DNA gyrase	[147]
Emodin	C_15_H_10_O_5_	3220	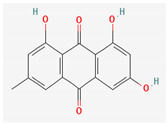	*Rheum palmatum* L.	Methicillin-resistant *Staphylococcus aureus*, vancomycin-resistant *Enterococcus faecium*	Inhibition of DNA gyrase	[157]
Curcumin	C_21_H_20_O_6_	969516	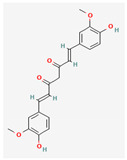	*Curcuma longa* L.	*Staphylococcus aureus*, *Escherichia coli.*	Enhancing membrane permeability, inhibition of enzyme sortase A	[155]
Quercetin	C_15_H_10_O_7_	5280343	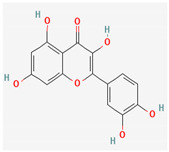	*Vitaceae* Juss. *Brassicaceae* Burnett, *Amaryllidaceae* J.St.-Hil., *Rutaceae* Juss.	*Staphylococcus aureus*, *Escherichia coli*, *Helicobacter pylori*	Efflux pump inhibitor, inhibition of d-alanine:d-alanine ligase in *Helicobacter pylori* and *Escherichia coli*	[148,158]
Kaempferol	C_15_H_10_O_6_	5280863	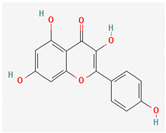	*Alpinia calcarata*–Roscoe.	Methicillin-resistant *Staphylococcus aureus*, *Candida albicans*	Efflux pump inhibitor	[150,151]
Resveratrol	C_14_H_12_O_3_	445154	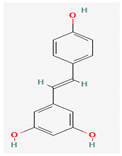	*Vitis vinifera* L.	*Campylobacter jejuni*	Efflux pump inhibitor	[159]
Taxifolin/dihydroquercetin	C_15_H_12_O_7_	439533	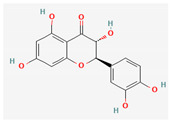	Conifers-like *Larix sibirica* Ledeb., *Pinus roxburghii* Sarg., *Cedrus deodara* (Roxb. ex D.Don) G.Don, *Taxus chinensis* (Pilg.) Rehder.	Methicillin-resistant *Staphylococcus aureus*, *Enterococcus faecalis*	Cysteine transpeptidase sortase A (SrtA) inhibitor, β-ketoacyl acyl carrier protein synthase inhibitor	[160,161]
Osthole	C_15_H_16_O_3_	10228	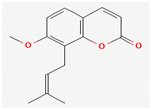	*Prangos hulusii (*Şenol, Yıldırım & Seçmen), *Cnidium monnieri* (L.) Cusson ex Juss., *Angelica pubescens* Maxim.	*Bacillus subtilis*, *Staphylococcus aureus*, *Klebsiella pneumoniae*	DNA gyrase inhibitor, MCR-1 inhibitor	[162,163]
Galbanic acid	C_24_H_30_O_5_	7082474	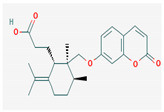	*Ferula szowitsiana* DC.	*Staphylococcus aureus*	Efflux pump inhibitor	[164]
Asphodelin A	C_15_H_10_O_6_	54679752	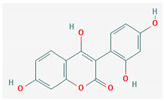	*Asphodelus microcarpus* Rchb.	*Staphylococcus aureus*, *Escherichia coli*, *Pseudomonas aeruginosa*, *Candida albicans*, *Botrytis cinerea*	DNA gyrase inhibitor	[165]
Aegelinol	C_14_H_14_O_4_	600671	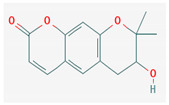	*Phlojodicarpus villosus* (Turcz. ex Fisch. & C.A.Mey.) Turcz. ex Ledeb., *Peucedanum praeruptorum* Dunn, *Ferulago galbanifera* (Mill.) W.D.J.Koch	*Salmonella enterica serovar typhi*, *Enterobacter aerogenes*, *Enterobacter cloacae*, *Staphylococcus aureus*	DNA gyrase inhibitor	[92,166]
3,4,5-trihydroxybenzoic acid (Gallic acid)	C_7_H_6_O_5_	370	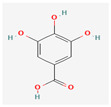	*Mimosa bimucronata* (DC.) Kuntze, *Punica granatum* L.	*Staphylococcus aureus*, *Escherichia coli*, *Listeria monocytogenes*, *Pseudomonas aeruginosa*	Cell membrane disintegration	[145]
Ferulic acid	C_10_H_10_O_4_	445858	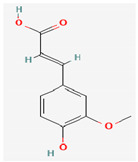	*Commelinaceae* Mirb.	*Staphylococcus aureus*, *Escherichia coli*, *Listeria monocytogenes, Pseudomonas aeruginosa*	Cell membrane disintegration	[145]
Apigenin	C_15_H_10_O_5_	5280443	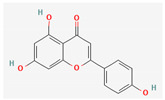	*Matricaria chamomilla* L.	*Pseudomonas aeruginosa*	NA	[134,148]
Genistein	C_15_H_10_O_5_	5280961	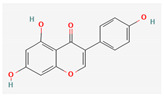	*Glycine max* (L.) Merr.	*Pseudomonas aeruginosa*	NA	[167]
Eriodictyol	C_15_H_12_O_6_	440735	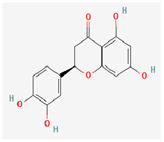	*Eriodictyon californicum* (Hook. & Arn.) Decne.	*Enterococcus* *faecalis*	NA	[161]
Agasyllin	C_19_H_20_O_5_	15596603	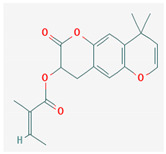	*Ferulago galbanifera* (Mill.) W.D.J.Koch	*Campylobacter* species	DNA gyrase inhibitor	[166]

* Chemical structures of compounds have been taken from PubChem; www.pubchem.ncbi.nlm.nih.gov (Accessed on 10 December 2022).

**Table 5 pharmaceuticals-16-00881-t005:** List of important organosulfur/isothiocyanate classes of plant-based antimicrobial agents reported and their sources, pathogenic targets, and mechanisms of action.

Bioactive Compound	Chemical Formula	PubChem CID	Chemical Structure *	Plant Source	Target Pathogen	Mode of Action	References
Allicin	C_6_H_10_OS_2_	65036	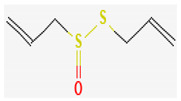	*Allium sativum* L.	*Salmonella typhimurium*, *Staphylococcus aureus*, *Bacillus subtilis*, *Bacillus typhosus*, *Bacillus paratyphosus*, *Morganella* *morganii*, *Bacillus enteritidis*, *Shigella dysenteriae*, *Vibrio cholera*, *Escherichia. coli*, *Listeria monocytogenes*, *Helicobacter pylori*, drug-resistant strains of *Mycobacterium tuberculosis*	Sulfhydryl-dependent enzyme inhibitor, DNA/RNA synthesis inhibitor, inhibitor of acetyl-CoA synthases in yeasts	[169,174,178,179]
Ajoene	C_9_H_14_OS_3_	5386591	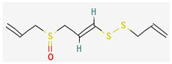	*Allium sativum* L.	*Campylobacter jejuni*, *Staphylococcus aureus*, *Escherichia coli*, *Helicobacter pylori*	Sulfhydryl-dependent enzyme	[168,180]
Sulforaphane	C_6_H_11_NOS_2_	5350	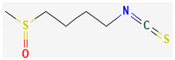	*Brassicaceae* Burnett.	*Bacillus cereus*, *Escherichia coli*	Membrane destruction, ATP synthase inhibitor, DNA/protein synthesis inhibitor	[181]
Allyl isothiocyanates (AITCs)	C_4_H_5_NS	5971	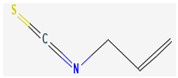	*Armoracia rusticana* G.Gaertn., B.Mey., & Scherb.	Oral pathogens, *Helicobacter pylori, Escherichia coli*	Inhibition of urease, reducing the inflammatory component of *Helicobacter* infections, inhibition of ATP binding sites of P-ATPase in bacteria	[92,170]
Benzyl isothiocyanate (BITC)	C_8_H_7_NS	2346	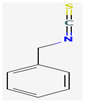	*Alliaria petiolata* (M.Bieb.) Cavara & Grande	Methicillin-resistant *Staphylococcus aureus*	Disruption of membrane integrity	[176]
Phenethyl isothiocyanate(PEITC)	C_9_H_9_NS	16741	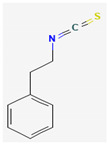	*Brassica campestris* L., *Brassica rapa* L.	Gram-positive bacteria	Intracellular accumulation of reactive oxygen species (ROS), depolarization of mitochondrial membrane	[92,182]
Berteroin	C_7_H_13_NS_2_	206037	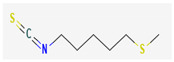	*Brassica oleracea* L.	*Bacillus subtilis*, *Escherichia coli*, *Helicobacter pylori*	NA	[93]
Cheirolin	C_5_H_9_NO_2_S_2_	10454	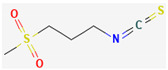	*Cheiranthus cheiri* L.	*Helicobacter pylori*	NA	[134]
Alyssin	C_7_H_13_NOS_2_	206035	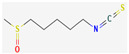	*Alyssum* L. sp.	*Helicobacter pylori*	NA	[134]

* Chemical structures of compounds have been taken from PubChem; www.pubchem.ncbi.nlm.nih.gov (Accessed on 10 December 2022).

**Table 6 pharmaceuticals-16-00881-t006:** List of important terpene classes of plant-based antimicrobial agents reported and their sources, pathogenic targets, and mechanisms of action.

Bioactive Compound	Chemical Formula	PubChem CID	Chemical Structure *	Plant Source	Target Pathogen	Mode of Action	References
Eugenol	C_10_H_12_O_2_	3314	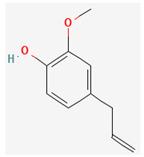	*Syzygium aromaticum* (L.) Merr. & L.M.Perr, *Cinnamomum zeylanicum* Blume.	*Helicobacter pylori*, Methicillin-resistant *Staphylococcus aureus*, Methicillin-sensitive *Staphylococcus aureus*, *Pseudomonas aeruginosa*	Inhibits biofilm construction, interrupts cell-to-cell communication, eradicates the pre-established biofilms, and kills the bacteria in biofilms	[190,192]
Cinnamaldehyde	C_9_H_8_O	637511	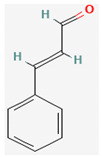	*Cinnamomum verum* J. Presl.	*Escherichia coli*, *Staphylococcus aureus*	Membrane damage	[193]
Ursolic acid	C_30_H_48_O_3_	64945	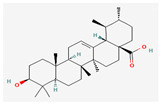	*Salvia rosmarinus* Spenn., *Salvia officinalis* L.	*Escherichia coli*	Cell membrane disturbance	[189]
Farnesol	C_15_H_26_O	445070	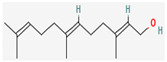	*Vachellia farnesiana* (L.) Wight & Arn.	*Staphylococcus aureus* including MRSA	Membrane damage	[194]
Carvacrol	C_10_H_14_O	10364	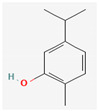	*Thymus capitatus* (L.) Hoffmanns. & Link., *Thymus vulgaris* L.	*Escherichia coli*	Cell membrane damage	[167,195]
Nerolidol	C_15_H_26_O	5284507	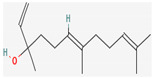	*Cannabis sativa* L.	*Staphylococcus aureus* including MRSA	Cell membrane damage	[194]
Thymol	C_10_H_14_O	6989	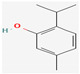	*Thymus capitatus* (L.) Hoffmanns. & Link.	*Staphylococcus aureus* including MRSA	NA	[195,196]

* Chemical structures of compounds have been taken from PubChem; www.pubchem.ncbi.nlm.nih.gov (Accessed on 10 December 2022).

**Table 7 pharmaceuticals-16-00881-t007:** List of plants found in the Indian Himalayan region and their reported antimicrobial bioactive compounds.

Plant Name	Bioactive Compounds	Target Pathogen	References
*Abrus precatorius* L.	6-propionyloxymethyl-4′,5,7-trihydroxyisoflavanone	*Bacillus cereus*, *Escherichia coli*	[223]
*Abutilon theophrasti* Medik.	Rutin,	*Salmonella enterica*, *Escherichia coli*, *Streptococcus pneumoniae*, *Staphylococcus aureus*	[224]
	quercetin 7-o-β-glucoside,		
	kaempferol 3-o-α-rhamnopyranosyl (1→6)-β-glucopyranoside,		
	luteolin,		
	apigenin 7-o-β-diglucoside,		
	poncirin,		
	tiliroside		
*Achillea millefolium* L.	Camphor, germacrene-d, (e)-nerolidol, sabinene, (e)-p-mentha-2,8-dien-1-ol, 1,8-cineole	*Salmonella typhimurium*, *Salmonella agona*	[225]
*Achyranthes aspera* L.	Achyranthine, betaine, betanin, isobetanin	*Bacillus subtilis*, *Escherichia coli*, *Pseudomonas aeruginosa*, *Staphylococcus aureus*	[226]
*Aconitum* *violaceum* Jacq. ex Stapf.	Ethyl acetate fraction	*Escherichia coli*, *Shigella flexneri*, *Bacillus subtilis*, *Staphylococcus aureus*	[227]
*Aconitum* *heterophyllum* Wall. ex Royle	6-dehydroacetylsepaconitine,	*Staphylococcus aureus*, *Salmonella typhi*, *Pseudomonas aeruginosa*	[228]
	13-hydroxylappaconitine, lycoctonine,		
	lappaconitine		
*Acorus calamus* L.	Asarone	*Aspergillus niger*, *Candida albicans*	[229]
*Adiantum* *capillus*-*veneris* L.	3-p-coumaroylquinic acid, kaempferol 3-o-glucoside	*Staphylococcus aureus*, *Staphylococcus epidermidis*, β-hemolytic *Streptococcus*, *Enterococcus faecalis*, *Escherichia coli*, *Pseudomonas aeruginosa*	[230]
*Adiantum* *pedatum*	Ethyl and acetone extracts	*Staphylococcus aureus*, *Klebsiella pneumoniae*, *Pseudomonas aeruginosa*, *Escherichia coli*	[231]
*Aegle* *marmelos* (L.) Corrêa	Limonene, β-ocimene, germacrene, α-phellandrene	*Caenorhabditis elegans*	[232]
*Ageratum houstonianum* Mill.	Ageratochromene, demothoxyageratochromene, β-caryophyllene	*Micrococcus luteus*, *Rhodococcus rhodochrous*	[233]
*Ajuga parviflora* Benth.	Ajugin A, ajugin B	*Citrobacter* sp., *Pseudomonas aeruginosa*	[234]
*Allamanda* *cathartica* L.	Silver nanoparticles of flower aqueous extracts	*Salmonella typhimurium*, *Staphylococcus aureus*, *Escherichia coli*, *Klebsiella pneumoniae*	[235]
*Allium* *cepa* L.	Allicin	*Salmonella typhimurium*, *Staphylococcus aureus*, *Escherichia coli*	[236]
*Allium* *sativum* L.	Allicin,	*Aspergillus versicolor*, *Penicillium citrinum*, *Burkholderia cepacia*, *Staphylococcus aureus*, *Escherichia coli*, *Bacillus subtilis*, *Penicillium funiculosum*, *Candida albicans*, *Helicobacter pylori*	[179]
	diallyl sulfide,		
	diallyl disulfide,		
	diallyl trisulfide,		
	e/z-ajoene,		
	s-allyl-cysteine,		
	s-allyl-cysteine sulfoxide.		
*Amaranthus* *caudatus* L.	Ferulic acid	*Escherichia coli*	[145]
*Amaranthus viridis* L.	Rutin, quercetin, spinosterol, amasterol	*Staphylococcus aureus*, *Escherichia coli*, *c*, *Rhizopus oligosporus*, *Colletotrichum musae*, *Fusarium solani*	[237]
*Amomum subulatum* Roxb.	1,8-cineole, α-terpineol, α-pinene, β-pinene	*Aspergillus niger*	[238]
*Angelica glauca* Edgew.	β-phellandrene, (z)-ligustilide methyl octane, limonene, β-phellandrene, β-pinene, (z)-3-butylidene-phthalide, (z)-ligustilide, (e)-ligustilide, citronellyl acetate	*Clostridium difficile*, *Clostridium perfringens*, *Enterococcus faecalis*, *Eubacterium limosum*, *Peptostreptococcus anaerobius*, *Candida albicans*	[239]
*Arctium* *lappa* L.	Chlorogenic acid, caffeic acids	*Pseudomonas aeruginosa*, *Bacillus cereus*	[240]
*Arnebia* *benthamii* (Wall. ex G.Don) I.M.Johnst.	Shikonin,	*Escherichia coli*, *Pseudomonas aeruginosa*, *Shigella flexneri*, *Klebsiella pneumoniae*, *Salmonella typhimurium*, *Staphylococcus aureus*	[241]
	alkanin hoslundal,		
	artemidiol,		
	ganoderiol,		
	2-hexaprenyl-6-hydroxyphenol		
*Artemisia* *dubia* Wall. ex Besser	Chrysanthenone, coumarin, camphor	*Aspergillus niger*	[242]
*Artemisia indica* Willd.	Isoascaridole, trans-p-mentha-2,8-dien-1-ol, trans-verbenol, artemisia ketone, germacrene B, borneol, cis-chrysanthenyl acetate, davanone, β-pinene.	*Bacillus subtilis*, *Staphylococcus epidermidis*, *Pseudomonas aeruginosa*, *Salmonella typhi*, *Klebsiella pneumoniae*, *Penicillium chrysogenum*, *Aspergillus niger*	[242,243]
*Asparagus* *racemosus* Willd.	Catecholic tannin, saponin, gallic tannin	*Escherichia coli*, *Salmonella typhimurium*, *Bacillus subtilis*, *Pseudomonas aeruginosa*, *Staphylococcus aureus*, *Klebsiella pneumoniae*, *Enterococcus faecalis*, *Saccharomyces cerevisiae*	[244]
*Atropa* *acuminata* Royle ex Lindl.	Aqueous extract	*Bacillus Subtilis*, *Escherichia coli*, *Klebsiella pneumoniae*, *Pseudomonas aeruginosa*, *Salmonella typhimurium*, *Staphylococcus aureus*	[245]
*Atropa* *bella-donna* L.	Ethanolic extracts	*Staphylococcus aureus*, *Escherichia coli*	[246]
*Bacopa* *monnieri* (L.) Wettst.	Luteolin	*Staphylococcus aureus*, *Alternaria alternate*, *Fusarium acuminatum*	[247]
*Baliospermum* *montanum* (Willd.) Müll.Arg.	Leaf (methanolic and aqueous extract), callus (acetone and ethanolic extract)	*Bacillus subtilis*, *Klebsiella pneumoniae*, *Staphylococcus aureus*, *Escherichia coli*	[248]
*Berberis lyceum* Royle	Berberine	*Streptococcus agalactiae*, *Staphylococcus aureus*, *Streptococcus mutans*, *Streptococcus pyogenes*, *Corynebacterium diphtheriae*	[110]
*Bergenia* *ciliate* (Haw.) Sternb.	Pyrogallol,	*Staphylococcus aureus*, *Bacillus subtilis*, *Bacillus megaterium*, *Escherichia coli*, *Serratia marcescens*, *Nocardia tenerifensis*, *Streptomyces* sp., *Aspergillus niger*, *Fusarium oxysporum*	[249]
	rutin,		
	morin,		
	bergenin,		
	catechin,		
	gallic acid		
*Betula utilis* D.Don	Geranic acid,	*Staphylococcus aureus*, *Bacillus subtilis*, *Pseudomonas aeruginosa*, *Escherichia coli*	[250]
	β-seleneol,		
	β-linalool,		
	β-sesquiphellendrene,		
	champacol,		
	1,8-cineol.		
*Bidens* *biternate* (Lour.) Merr. & Sherff	Methanolic extract	*Escherichia coli*, *Klebsiella pneumoniae*, *Pseudomonas* sp., *Staphylococcus aureus*, *Staphylococcus epidermidis*	[251]
*Blumea lacera* (Burm.f.) DC.	Lachnophyllum ester, lachnophyllic acid, germacrene d, β-farnesene.	*Staphylococcus aureus*, *Candida albicans*, *Aspergillus niger*	[252]
*Bridelia* *retusa* (L.) A.Juss.	Ethanolic extract	*Pseudomonas aeruginosa*, *Escherichia coli*	[253]
*Calendula* *officinalis* L.	Selenium nanoparticles of methanolic extract of flowers	*Serratia marcescens*, *Enterobacter cloacae*, *Alcaligenes faecalis*	[254]
*Calotropis procera* (Aiton) W.T.Aiton	α-amyrin, lupeol acetate, phytol, hexadecanoic acid, stigmasterol, linolenic acid, gombasterol A	*Staphylococcus aureus*, *Klebsiella pneumoniae*, *Escherichia coli*	[255]
*Caltha* *palustris* L.	Methanolic extract	*Staphylococcus epidermidis*, *Proteus vulgaris*	[256]
*Cannabis* *sativa* L.	Cannabidiol (cannabinoids)	*Staphylococcus aureus (MDR*, *MRSA)*, *Staphylococcus epidermidis*, *Streptococcus pneumoniae*, *Streptococcus pyogenes*, *Enterococcus faecium*, *Cutibacterium acnes*, *Clostridium difficile*, *Escherichia coli*, *Klebsiella pneumoniae*, *Pseudomonas aeruginosa*, *Acinetobacter baumannii*, *Serratia marcescens*, *Proteus mirabilis*, *Salmonella typhimurium*	[257]
*Cassia fistula* L.	Eugenol, phytol, camphor, linonene, salicyl alcohol, 4-hydroxybenzyl alcohol	*Aspergillus niger*, *Candida albicans*	[229]
*Cassia tora* L.	Elemol, linalool, palmitic acid	*Bacillus cereus*, *Staphylococcus aureus*	[229]
*Cedrus deodara* (Roxb. ex D.Don) G.Don	Wikstromal, matairesinol, dibenzylbutyrolactol, berating, isopimpillin, lignans 1, 4 diaryl butane, benzofuranoid neo lingam, isohemacholone, sesquiterpenes, deodarone, atlantone, deodarin, deodardione, limonenecarboxylic acid, α-himacholone, β-himacholone, α-pinene, β-pinene, myrcene, cedrin (6-methyldihydromyricetin), taxifolin, cedeodarin (6-methyltaxifolin), dihydromyricetin and cedrinoside	*Escherichia coli*	[258]
*Chaerophyllum villosum* Wall. ex DC.	γ-terpinene, p-cymene, carvacrol methyl ether, myristicin, thymol	*Staphylococcus aureus*, *Streptococcus mutans*, *Candida albicans*, *Candida glabrata*	[259]
*Chenopodium* *ambrosioides* L.	Rutin (3,3′,4′,5,7-pentahydroxyflavone-3-rhamnoglucoside)	*Staphylococcus aureus*, *Pseudomonas aeruginosa*, *Enterococcus faecalis*, *Paenibacillus apiarius*, *Paenibacillus thiaminolyticus*	[260]
*Cichorium* *intybus* L.	Triterpenois, cichoridiol, intybusoloid, lupeol, fridelin, β- sitosterol, sigmasterol, betulinic acid, betunaldehyde, syringic acid, vanilic acid	*Pseudomonas aeruginosa*, *Staphylococcus aureus*	[260]
*Cinnamomum glanduliferum* (Wall.) Meisn.	1,8-cineole, α-pinene, α-terpineol, germacrene d-4-ol, α-thujene	*Micrococcus luteus*, *Escherichia coli*, *Pseudomonas aeruginosa*, *Aeromonas salmonicida*	[261]
*Cissampelos pareira* L.	Bis-benzylisoquinoline, benzylisoquinoline, tropoloisoquinoline, aporphine, azafluoranthene, protoberberine	*Staphylococcus aureus*, *Streptococcus pneumoniae*, *Escherichia coli*, *Pseudomonas aeruginosa*, *Klebsiella pneumoniae*, *Proteus vulgaris*	[262]
*Convolvulus* *arvensis* L.	Butanolic extract	*Staphylococcus aureus*, *Acinetobacter junii*, *Klebsiella pneumoniae*, *Acinetobacter baumannii*, *Escherichia coli*, *Enterococcus faecalis*, *Pseudomonas aeruginosa*, *Salmonella dysenteriae*, *Vibrio cholera*, *Proteus mirabilis*, *Salmonella paratyphi*, *Serratia marcescens*, *Enterobacter cloacae*	[263]
*Coriandrum* *sativum* L.	β-linalool (essential oil)	*Bacillus subtilis*, *Stenotrophomonas maltophilia*	[264]
*Crocus sativus* L.	Crocin, safranal semi-synthetic safranal derivatives	*Helicobacter pylori*, *Staphylococcus aureus*, *Listeria* spp. *Bacillus subtilis*, *Bacillus cereus*, *Salmonella enterica*, *Shigella dysenteriae*, *Escherichia coli*	[265]
*Curcuma Longa* L.	α-turmerone, β-turmerone, α-phellandrene, 1,8-cineole, *p*-cymene, terpinolene	*Bacillus cereus*, *Staphylococcus aureus*, *Aspergillus niger*	[266]
*Cuscuta reflexa* Roxb.	cis-3-butyl-4-vinylcyclopentane, limonene, (e)-nerolidol	*Aspergillus niger*	[267]
*Cymbopogon* *citratus* (DC.) Stapf	α-citral,	*Pseudomonas aeruginosa*, *Escherichia coli*, *Klebsiella pneumoniae*, *Staphylococcus aureus*, *Bacillus subtilis*	[268]
	β-citral,		
	myrcene		
*Cyperus* *rotundus* L.	α.-pinene, camphene, D-limonene,	*Bacillus subtilis*, *Pseudomonas aeruginosa*, *Escherichia coli*, *Staphylococcus aureus* (ARSA)	[269,270]
	camphenol, p-mentha-1,5-dien-8-ol,		
	thymol, myrtenal, carveol, copaene,		
	caryophyllene, naphthalene,		
	1,6-dimethyl-4-(1-methylethyl).		
*Datura metel* L.	Daturaolone	*Escherichia coli*, *Staphylococcus aureus*, *Bacillus subtilis*, *Klebsiella pneumoniae*, *Staphylococcus epidermidis*	[271]
*Datura* *stramonium* L.	Chloroform extracts	*Staphylococcus aureus*	[272]
*Daucus* *carota* L.	Methylisoeugenol	*Campylobacter jejuni*	[273]
*Dioscorea* *bulbifera* L.	Bafoudiosbulbins, 2,7-dihydroxy-4-methoxyphenanthrene	*Escherichia coli*, *Enterobacter aerogenes*, *Klebsiella pneumoniae*, *Pseudomonas aeruginosa*	[274]
*Dodecadenia grandiflora* Nees	Germacrene D, furanodiene	*Staphylococcus aureus*, *Pasteurella multocida*	[275]
*Dodonaea viscosa* Jacq.	Hautriwaic acid, dodonoside B, dodonic acid, kaempferol, sakuranetin, dehydrohautriwaic acid, hautriwaic lactone, alizarin, penduletin, 3,5,7-trihydroxy-4′-methoxyflavone, isorhamnetin-3-rhamnosylgalactoside, donoside a, 5- hydroxy-3,6,7,4′-tetra methoxy flavone	*Streptococcus pyogenes*, *Escherichia coli*, *Klebsiella pneumonia*, *Pseudomonas fluorescens*, *Staphylococcus aureus*, *Bacillus subtilis*	[276]
*Epimedium grandiflorum* C. Morren	Hydroethanolic extract	*Bacillus subtilis*, *Staphylococcus aureus*, *Escherichia coli*, *Klebsiella pneumoniae*, *Acinetobacter* sp., *Pseudomonas* sp., *Salmonella* sp.	[277]
*Equisetum* *diffusum* D. Don	Aqueous extract	*Escherichia coli*, *Micrococcus luteus*, *Pseudomonas aeruginosa*, *Bacillus pumilus*, *Bacillus cereus*, *Bacillus licheniformis*, *Salmonella typhi*, *Streptococcus mutans*	[278]
*Eupatorium adenophorum* Spreng.	*p*-cymene, bornyl acetate, amorph-4-en-7-ol, camphene	*Arthrobacter protophormiae*, *Escherichia coli*, *Micrococcus luteus*, *Rhodococcus rhodochrous*, *Staphylococcus aureus*	[233]
*Euphorbia* *helioscopia* L.	Euphoheliosnoid E	*Streptococcus mutans*, *Actinomyces viscosus*	[279]
*Euphorbia* *wallichii* Hook. f.	Acorenone B, cycloisosativene, β-cedrene, copaene, 3β-hydroxy-5α-androstane, palmitic acid	*Staphylococcus aureus*	[279]
*Foeniculum* *vulgare* Mill.	Dillapional	*Bacillus subtilis*	[280]
*Fritillaria* *roylei* Hook.	Peonidin	*Escherichia coli*, *Klebsiella pneumoniae*, *Micrococcus luteus*, *Staphylococcus pneumonia*, *Haemophilus influenza*, *Neisseria mucosa*	[281]
*Fumaria indica* (Hausskn.) Pugsley	n-octacosan-7β-ol	*Leishmania donovani promastigotes*, *Staphylococcus epidermidis*, *Escherichia coli*, *Candida albicans*, *Aspergillus niger*	[282]
*Galium aparine* L.	Chlorogenic acid, p-coumaric acid, ferulic acid, luteolin, rutin	*Staphylococcus aureus*, *Listeria monocytogenes*	[283]
*Gentiana kurroo* Royle	Flavonoids and phenols	*Proteus mirabilis*, *Streptococcus faecalis*, *Escherichia coli*, *Salmonella enteritidis*, *Micrococcus luteus*, *Enterobacter cloacae*	[284]
*Geranium* *wallichianum* D.Don ex Sweet	Leaf extracts conjugated with zinc oxide nanoparticles	*Bacillus subtilis*, *Staphylococcus aureus*, *Pseudomonas aeruginosa*, *Escherichia coli*, *Klebsiella pneumoniae*	[285]
*Girardinia* *diversifolia* (Link) Friis	β-sitosterol, 3-hydroxystigmast-5-en-7-one, 7-hydroxysitosterol	*Bacillus pumilus*, *Escherichia coli*, *Staphylococcus aureus*	[286]
*Gaultheria fragrantissima* Wall.	Methyl salicylate	*Staphylococcus aureus*	[259]
*Hedychium* *spicatum* G.Lodd.	Hedychenone, spicatanol, 6-endo-hydroxycineole	*Shigella boydii*, *Shigella sonnei*, *Shigella flexneri*, *Bacillus cereus*, *Vibrio cholera*, *Escherichia coli*, *Staphylococcus aureus*, *Pseudomonas aeruginosa*, *Klebsiella pneumoniae*	[287,288]
*Holarrhena* *antidysenterica* Wall.	Conessine	*Acinetobacter baumannii*, *Pseudomonas aeruginosa*	[120]
*Hyoscyamus* *niger* L.	Non-alkaloidal seed extract	*Bacillus subtilis*, *Escherichia coli*, *Staphylococcus aureus*	[289]
*Hypericum* *perforatum* L.	Hypericin	Methicillin-resistant *Staphylococcus aureus*, methicillin-sensitive *Staphylococcus aureus*, *Escherichia coli*	[290]
*Inula cappa* (Buch.-Ham. ex D.Don) DC.	β-caryophyllene, *cis*-dihydro-mayurone, β-bisabolene, (E)-β-farnesene	*Enterococcus faecalis*, *Klebsiella pneumoniae*, *Xanthomonas phaseoli* and *Bacillus subtilis*	[291]
*Inula* *racemose* Hook.f.	Isoalantolactone	*Bacillus subtilis*, *Escherichia coli*, *Pseudomonas fluorescens*, *Staphylococcus lentus*, *Staphylococcus aureus*	[292]
*Iris ensata* Thunb.	Methanolic extracts	*Bacillus cereus*, *Pseudomonas aeruginosa*, *Proteus vulgaris*, *Escherichia coli*	[293]
*Iris kashmiriana* Baker	Irigenin, iridin, junipeginin-c,	*Bacillus subtilis*, *Staphylococcus epidermidis*, *Proteus vulgaris*, *Pseudomonas aeruginosa*, *Staphylococcus aureus*, *Staphylococcus typhimurium*, *Escherichia coli*, *Shigella dysenteriae*, *Klebsiella pneumoniae*	[293]
	resveratrol, piecid, resveratroloside,		
	isorhamnetin-3-oneohesperidoside		
*Iris* *nepalensis* D.Don	Methanolic extract	*Staphylococcus aureus*, *Escherichia coli*, *Pseudomonas aeruginosa*	[294]
*Jasminum* *officinale* L.	Ethanolic extract	Methicillin-resistant *Staphylococcus aureus*	[295]
*Juglans regia* L.	α-pinene, β-pinene, β-caryophyllene, germacrene d, limonene, eugenol, methyl salicylate, germacrene d, (*e*)-β-farnesene	*Bacillus subtilis*, *Staphylococcus epidermidis*, *Proteus vulgaris*, *Pseudomonas aeruginosa*, *Staphylococcus aureus*, *Salmonella typhi*, *Escherichia coli*, *Shigella dysenteriae*, *Klebsiella pneumoniae*	[296]
*Juniperus macropoda* Boiss.	Sabinene, terpinen-4-ol, cedrol, β-elemene, trans-sabinene hydrate, α-cubebene, α-thujone, biformene	*Candida albicans*, *Colletotrichum fragariae*, *Colletotrichum gloeosporioides*	[297]
*Lagenaria siceraria* (Molina) Standl.	β-carotene, 22-deoxocurcubitacin-d, 22-deoxoisocurcubitacin D, avenasterol, codisterol, elesterol, isofucasterol, stigmasterol, sitosterol, compesterol, spinasterol, 7-0-glucosyl-6-c-glucoside apigenin, 6-c-glucoside apigenin, 6-cglucoside luteolin, 7,4′-o-diglucosyl- 6-c-glucoside, apigenin	*Staphylococcus aureus*, *Pseudomonas* sp., *Escherichia coli*, *Bacillus subtilis*, *Candida* sp., *Aspergillus niger*	[298]
*Lantana camara* (Hayek) R.W.Sanders	Germacrene B, β-caryophyllen, 3,7,11-trimethyl-1,6,10-dodecatriene, β-caryophyllene, zingiberene, γ-curcumene, davanone, (E)-nerolidol	*Arthrobacter protophormiae*, *Micrococcus luteus*, *Rhodococcus rhodochrous*, *Staphylococcus aureus*	[233]
*Lavandula* *stoechas* L.	1,8-cineole,	Methicillin-resistant *Staphylococcus aureus*, *Klebsiella pneumoniae*, *Salmonella typhimurium*	[299]
	fenchone,		
	camphor		
*Lindera neesiana* (Wall. ex Nees) Kurz	Geranial, neral, citronellal, 1,8-cineole, α-pinene, β-pinene, methyl chavicol, safrole	*Staphylococcus aureus*, *Candida albicans*	[300]
*Lindera pulcherrima* (Nees) Benth. ex Hook.f.	Curzerenone, furanodienone	*Staphylococcus aureus*, *Salmonella enterica*	[275]
*Mallotus philippensis* (Lam.) Müll.Arg.	Bergenin, mallotophilippinens, rottlerin, and isorottlerin	*Bacillus cereus* var mycoides, *Bacillus pumilus*, *Bacillus subtilis*, *Bordetella bronchiseptica*, *Micrococcus luteus*, *Staphylococcus aureus*, *Staphylococcus epidermidis*, *Escherichia coli*, *Klebsiella pneumoniae*, *Candida albicans*, *Saccharomyces cerevisiae*	[301,302]
*Malva neglecta* Wallr.	Hydrotyrosol, coumaroylhexoside, kaempferol-3-(p-coumaroyldiglucoside)-7-glucoside, quercetin-3-o-rutinoside, epicatechin-3-o-(4-o-methyl)-gallate, oleic acid, taurine, ethylene dimercaptan, isoeugenol, patchoulane, methyl 12-methyltetradecanoate, isopropyl myristate	*Escherichia coli*, *Staphylococcus aureus*, *Klebsiella pneumoniae*, *Salmonella typhi*, *Bacillus subtilis*, *Aspergillus fumigatus*, *Aspergillus flavus*, *Aspergillus niger*, *Fusarium solani*	[303]
*Marrubium* *vulgare* L.	Methanolic extract	*Escherichia coli*, *Bacillus subtilis*, *Staphylococcus aureus*, *Staphylococcus epidermidis*, *Pseudomonas aeruginosa*, *Proteus vulgaris*, *Candida albicans*	[304]
*Melia azedarach* L.	Crude extract	*Bacillus subtilis*, *Proteus mirabilis*, *Shigella flexneri*, *Proteus mirabilis*, *Shigella flexneri*, *Staphylococcus aureus*, *Bacillus subtilis*, *Pseudomonas aeruginosa*, *Shigella flexneri*	[305]
*Morina longifolia* Wall. ex DC.	Germacrene d, α-pinene, bicyclogermacrene, α-cadinol, (*e*)-citronellyl tiglate, β-phellandrene	*Escherichia coli*, *Staphylococcus aureus*, *Proteus vulgaris*, *Klebsiella pneumoniae*, *Bacillus subtilis*, *Pseudomonas aeruginosa*, *Alternaria alternata*, *Aspergillus flavus*, *Aspergillus fumigatus*, *Fusarium solani*	[306,307]
*Nepeta* *cataria* L.	Nepetalactone,	*Neisseria subflava*, *Citrobacter freundii*, *Branhamella ovis*, *Aeromonas caviae*, *Escherichia coli*, *Serratia marcescens*, *Enterococcus species*, *Staphylococcus aureus*	[308,309]
	β-caryophyllene,		
	thymol		
*Nardostachys jatamansi* (D.Don) DC.	β-gurjunene, valerena-4,7(11)-diene (7.1%), nardol a, 1(10)-aristolen-9β-ol, jatamansone	*Bacillus cereus*, *Escherichia coli*, *Candida albicans*	[310]
*Oxalis* *corniculate* L.	Methanolic extract	*Staphylococcus aureus*, *Escherichia coli*, *Shigella dysenteriae*, *Shigella flexneri*, *Shigella boydii*, *Shigella sonnei*	[311]
*Paeonia emodi* Royle	Leaf extract nanoparticles	*Staphylococcus aureus*, *Bacillus subtilis*, *Escherichia coli*, *Salmonella typhi*, *Pseudomonas aeruginosa*, *Klebsiella pneumoniae*	[312]
*Phoebe lanceolata* (Nees) Nees	1,8-cineole, β-caryophyllene	*Escherichia coli*	[275]
*Persicaria hydropiper* (L.) Delarbre	Confertifolin, polygodial	*Enterococcus faecalis*, *Bacillus subtilis*, *Staphylococcus aureus*, *Escherichia coli*, *Salmonella enterica*, *Epidermophyton floccosum*, *Curvularia lunata*, *Scopulariopsis* sp., *Candida albicans*, *Candida utilis*, *Candida krusei*, *Cryptococcus neoformans*, *Saccharomyces cerevisiae*, *Epidermophyton floccosum*, *Trichophyton mentagrophytes*, *Penicillium marneffei*	[313]
*Plantago lanceolata* L.	Luteolin 7-glucoside, hispidulin 7-glucuronide, luteolin 7-diglucoside, apigenin 7-glucoside, nepetin 7-glucoside and luteolin 6-hydroxy 4′-methoxy 7-galactoside, oleanolic acid, sitosterol acid, 18β-glycyrrhetinic, plantamajoside, verbacoside, 10-hydroxymajoroside, 10-acetoxymajoroside	*Bacillus subtilis*, *Staphylococcus aureus*, *Candida albicans*, *Candida tropicalis*, *Escherichia coli*, *Streptococcus pneumoniae*	[314]
*Podophyllum hexandrum* Royle	Phthalic acid,	*Bacillus megaterium*, *Pseudomonas aeruginosa*, *Aspergillus flavus*, *Fusarium solani*, *Staphylococcus aureus*, *Salmonella typhi*, *Klebsiella pneumoniae*, *Enterococcus faecalis*	[315,316]
	di-isobutyl ester,		
	1,2-benzenedicarboxylic acid,		
	diisooctyl ester,		
	polyneuridine,		
	podophyllotoxin,		
	β-sitosterol		
*Punica* *granatum* L.	Punicalagin	*Pseudomonas aeruginosa*, *Salmonella enteritidis*, *Escherichia coli*, *Staphylococcus epidermidis*, *Staphylococcus xylosus*, *Staphylococcus aureus*, *Bacillus cereus*, *Enterococcus faecium*, *Enterococcus faecalis*	[317]
*Prunus* *domestica* L.	Quercetin-3-o-galactoside	*Campylobacter jejuni*, *Salmonella typhimurium*, *Escherichia coli*, *Staphylococcus aureus*, *Listeria monocytogenes*	[318]
*Rheum* *emodi* Wall.	Emodin, rhein, chrysophanol dimethyl ether, resveratrol, revandchinone-4	*Escherichia coli*, *Staphylococcus aureus*, *Klebsiella pneumoniae*, *Bacillus subtilis*, *Pseudomonas aeruginosa*, *Klebsiella aerogenes*, *Bacillus sphaericus*, *Chromobacterium violaceum*	[319]
*Rhododendron anthopogon* D.Don	α-pinene, β-pinene, limonene, δ-cadinene	*Bacillus subtilis*, *Mycobacterium tuberculosis*, *Candida pseudotropicalis*	[320]
*Rumex dentatus* L.	Musizin, torachrysone-glucoside, 2-methoxystypandrone	*Escherichia coli*, *Klebsiella pneumoniae*, *Salmonella typhi*, *Pseudomonas aeruginosa*, *Bacillus subtilis*, *Streptococcus pneumoniae*, *Listeria monocytogenes*, *Staphylococcus epidermidis*, *Staphylococcus aureus*, *Bacillus cereus*	[321]
*Salix alba* L.	Anthocyanins, p-hydroxybenzoic, gallic acid, gentisic acid, sisymbrifolin, catechol	*Pseudomonas aeruginosa*, *Escherichia coli*, *Salmonella enterica*, *Staphylococcus aureus*	[322]
*Salvia sclarea* L.	Essential oil	*Escherichia coli*, *Staphylococcus aureus*, Methicillin-resistant *Staphylococcus epidermidis*	[323]
*Sambucus wightiana* Wall. ex Wight & Arn.	Gold nanoparticles of whole-plant extract	*Escherichia coli*, *Staphylococcus epidermidis*, *Salmonella enteritidis*	[324]
*Saussurea lappa* (Decne.) Sch.Bip.	Sesquiterpene lactones, zinc oxide nanoparticles of rhizome methanolic extract	*Staphylococcus aureus*, *Sphingobacterium thalpophilum*, *Staphylococcus aureus*, *Escherichia coli*, *Pseudomonas aeruginosa*, *Sphingobacterium* sp., *Acinetobacter* sp., *Ochrobactrum* sp.	[325]
*Skimmia laureola* (DC.) Decne.	Linalyl acetate, linalool, limonene, α-terpineol, geranyl acetate	*Staphylococcus aureus*, *Staphylococcus epidermidis*, *Aspergillus niger*, *Penicillium chrysogenum*	[326]
*Solanum tuberosum* L.	Potide-g, afp-j, potamin-1 or pg-2	*Staphylococcus aureus*, *Listeria monocytogenes*, *Escherichia coli*, *Candida albicans*	[327]
*Sonchus arvensis* L.	Phenols and flavonoids	*Escherichia coli*, *Salmonella enterica*, *Vibrio parahaemolyticus*, *Staphylococcus aureus*	[328]
*Stephania glabra* (Roxb.) Miers	Glabradine	*Staphylococcus aureus*, *Streptococcus. mutans*, *Microsporum gypseum*, *Microsporum canis*, *Trichophyton rubrum*	[329]
*Taraxacum officinale* F.H.Wigg.	9-hydroxyoctadecatrienoic acid, 9-hydroxyoctadecadienoic acid, vanillin, coniferaldehyde, p-methoxyphenylglyoxylic acid	*Staphylococcus aureus*, Methicillin-resistant *Staphylococcus aureus*, *Bacillus cereus*	[330]
*Terminalia arjuna* (Roxb. ex DC.) Wight & Arn.	Silver nanoparticles of bark extract	*Escherichia coli*	[331]
*Terminalia chebula* Retz.	1,2,6-tri-o-galloyl-β-d-glucopyranose	*Escherichia coli*, *Pseudomonas aeruginosa*, *Klebsiella pneumoniae*, *Staphylococcus aureus*, methicillin-resistant *Staphylococcus aureus*, *Proteus mirabilis*, *Acinetobacter baylyi*, *Bacillus megaterium*	[332,333,334]
*Valeriana jatamansi* (D.Don) Wall.	Maaliol, 3-methylvaleric acid, β-gurjunene	*Microsporum canis*, *Fusarium solani*	[335]
*Verbascum* *Thapsus* L.	1-hexzanol 2-hexene	*Klebsiella pneumoniae*, *Staphylococcus aureus*, *Escherichia coli*, *Mycobacterium phlei*, methicillin-resistant *Staphylococcus aureus*	[336]
*Viola* *odorata* L.	3-(2′,4′,6′,6′-tetramethylcyclohexa-1′,4′-dienyl) acrylic acid	*Haemophilus influenzae*, *Pseudomonas aeruginosa*, *Staphylococcus aureus*, *Streptococcus pneumoniae*	[337]
*Viscum* *album* L.	Hydroxycinnamic acids	*Xanthomonas campestris*, *Clavibacter michiganensis*, *Alternaria alternate*, *Fusarium oxysporum*	[338]
*Vitex negundo* L.	Methanolic extract	*Vibrio cholerae*, *Vibrio parahaemolyticus*, *Vibrio mimicus*, *Escherichia coli*, *Shigella* sp., *Aeromonas* sp.	[339]

## Data Availability

Data is contained within the article.

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
