# Peer review of "Phytochemicals as Antimicrobials: Prospecting Himalayan Medicinal Plants as Source of Alternate Medicine to Combat Antimicrobial Resistance"

_pharmaceuticals, 2023, doi:10.3390/ph16060881_

Round 1

Reviewer 1 Report

The authors have organized the manuscript in a very clear way. The manuscript is comprehensive and covers a wide range of Himalayan Medicinal Plants. However, the reliability of Google Scholar is questionable as some studies may not be peer reviewed. The authors need to elaborate on the this point, or exclude unreliable publications. 

The English is very good.

Reviewer 2 Report

The manuscript  ‘Phytochemicals as Antimicrobials: Prospecting Himalayan Medicinal Plants as Source of Alternate Medicine to Combat Antimicrobial Resistance’  describes the potential of plant-based antimicrobial to combat microbial resistance.

This overview helps to understand the mechanism of resistance to antibiotics and the diversity of natural plant based antimicrobials from Himalayan Medicinal Plants.

I have included some comments and some suggestions.

One of the comments I have is that the paper is too long which is not convenient for readers, therefore I advise to summarize some sections such 1.1. Timeline of antibiotic discovery: This section is not necessary.

The Abstract is Poorly written.

The authors start with ‘antimicrobial’ then ‘antibiotics’ it is confusing. Maybe it is necessary to start with identifying differences between antimicrobial and antibiotics. The antimicrobial is general terms includes antibiotics and other compounds …..

This review lacks information about the toxicity of phytochemicals, about the impact of climate change, about legislation and challenges of the use of phytochemicals in medicine.

Line 21: showed

Line 25: antibiotics or antimicrobial??

Line 50: reformulate

Line 55-56: reformulate

Line 237: 2.2.1.1.β-. Lactamases

Reviewer 3 Report

The manuscript ‘’Phytochemicals as Antimicrobials: Prospecting Himalayan Medicinal Plants as Source of Alternate Medicine to Combat Antimicrobial Resistance’’ reports interesting subject. It is well written. However, the paper needs to be deeply revised before acceptation for publication.

The introduction needs to be improved, the problematic is not well clear, authors are asked to reformulate some paragraphs by avoiding the well-known details.

Line 501: add authors of the scientific names. Please verify the scientific names in the whole manuscript

Line 86-96: I think the methodology must be in separate section.

Line 511: please define the synergistic interaction and clarify why it is the most preferable interaction in terms of antimicrobial therapies

Line 515: correct 1-4-napththoquinone

The manuscript ‘’Phytochemicals as Antimicrobials: Prospecting Himalayan Medicinal Plants as Source of Alternate Medicine to Combat Antimicrobial Resistance’’ reports interesting subject. It is well written. However, the paper needs to be deeply revised before acceptation for publication.

The introduction needs to be improved, the problematic is not well clear, authors are asked to reformulate some paragraphs by avoiding the well-known details.

Line 501: add authors of the scientific names. Please verify the scientific names in the whole manuscript

Line 86-96: I think the methodology must be in separate section.

Line 511: please define the synergistic interaction and clarify why it is the most preferable interaction in terms of antimicrobial therapies

Line 515: correct 1-4-napththoquinone

Round 2

Reviewer 1 Report

The manuscript in the current version is endorsed for publication